# Histone demethylase JMJD1C is phosphorylated by mTOR to activate de novo lipogenesis

Jose A. Viscarra[1,3], Yuhui Wang[1,3], Hai P. Nguyen[1], Yoon Gi Choi[2] & Hei Sook Sul[1]*

Fatty acid and triglyceride synthesis increases greatly in response to feeding and insulin. This lipogenic induction involves coordinate transcriptional activation of various enzymes in lipogenic pathway, including fatty acid synthase and glycerol-3-phosphate acyltransferase. Here, we show that JMJD1C is a specific histone demethylase for lipogenic gene transcription in liver. In response to feeding/insulin, JMJD1C is phosphorylated at T505 by mTOR complex to allow direct interaction with USF-1 for recruitment to lipogenic promoter regions. Thus, by demethylating H3K9me2, JMJD1C alters chromatin accessibility to allow transcription. Consequently, JMJD1C promotes lipogenesis in vivo to increase hepatic and plasma triglyceride levels, showing its role in metabolic adaption for activation of the lipogenic program in response to feeding/insulin, and its contribution to development of hepatosteatosis resulting in insulin resistance.

---

[1] Department of Nutritional Sciences & Toxicology, University of California, Berkeley, CA 94720, USA. [2] Functional Genomics Laboratory, University of California, Berkeley, CA 94720, USA. [3] These authors contributed equally: Jose A. Viscarra, Yuhui Wang. *email: hsul@berkeley.edu

Hepatosteatosis is a major metabolic disorder correlated with obesity and insulin resistance (IR). However, the relationship between them is still controversial: obesity and IR may cause hepatosteatosis, but whether hepatosteatosis drives the obesity-associated IR is not known[1]. Hepatosteatosis develops from a higher TG input than output and thus is influenced by de novo lipogenesis (DNL)[2]. DNL is attenuated during fasting, and increases drastically during feeding, with elevated circulating insulin and glucose[2]. Improved understanding of the regulation of lipogenesis may provide future therapeutic targets for the ever-increasing metabolic diseases linked to accumulation of excess fat.

Enzymes involved in lipogenesis, including fatty acid synthase (FAS), and mGPAT, are regulated coordinately at the transcriptional level[3,4]. Transcription factors (TF), including USF-1, ChREBP, LXR, and SREBP-1c, play critical roles in this process. USF-1 functions as a molecular switch by recruiting proteins to the lipogenic promoters[5]. In response to insulin, USF-1 interacts with DNA-PK and P/CAF for its phosphorylation and acetylation, respectively[6]. Phosphorylated and acetylated USF-1 then recruits SREBP-1c to bind its nearby site, and interacts and recruits BAF60c to bridge other BAF subunits to form LipoBAF complex at lipogenic genes for chromatin remodeling[7]. In addition, we found that a Mediator component, MED17 is phosphorylated by CK2 to be recruited by USF-1, thus linking USF-1 to the preinitiation complex formation for transcription[8]. Importantly, this common mechanism governs transcription of not only FAS but a battery of other lipogenic genes to fine-tune the rate of lipogenesis[4,9–11].

Epigenetic factors, such as histone modifiers, have been implicated in the development of metabolic disorders, such as hepatosteatosis and IR[12]. In adapting to environmental conditions, dynamic histone marks may affect gene transcription in a site- and tissue-specific fashion[13]. Although the best-studied histone modification so far is acetylation, histone methylation is an important modification in epigenetic regulation of gene expression and cell signaling[13,14]. The most common amino acid residue that can be methylated is lysine and its ε-amine group can undergo mono-, di- and trimethylation. For example, H3K4me3 is a well-accepted activating mark, whereas H3K9me3 represents a repressive mark[13]. There are many histone methyltransferases (HMT) and demethylases (HDM) that act on histones. These enzymes require co-factors or co-substrates. For example, JumonjiC domain-containing HDMs are 2-oxoglutarate-dependent oxygenases[15]. Thus, activities of histone modifiers are affected by intracellular metabolites and metabolic state of the cells[16]. It also is plausible that the histone modifiers may undergo posttranslational modification due to environmental cues, such as hormones and nutrients. However, the molecular details underlying the specificity of histone modifiers that function in specific processes or potential regulation of these enzymes by posttranslational modification have not been well studied.

Here, we identify JMJD1C of Jumonji family as a critical epigenetic factor for lipogenesis. We show that, by direct interaction with USF-1, JMJD1C is recruited to lipogenic promoters. We also show that JMJD1C is phosphorylated at T505 by mammalian target of rapamyci (mTOR) to be recruited to lipogenic genes in response to insulin/feeding. By ATAC-seq, chromatin immunoprecipitation sequencing (ChIP-seq) and RNA-seq, we show that JMJD1C is specific for lipogenesis by demethylating H3K9 altering the chromatin accessibility. JMJD1C has been identified in GWAS to be a candidate for aberrant serum triglyceride levels and type 2 diabetes[17–21]. Overall, JMJD1C plays a central epigenetic regulatory role for lipogenesis in response to insulin/feeding and its dysregulation contributes to development of hepatosteatosis and associated IR.

## Results

### USF-1 recruits JMJD1C to lipogenic promoters.
USF-1 is a central player as a molecular switch for lipogenesis during fasting/feeding by recruiting distinct TFs and signaling molecules[6,7]. To identify epigenetic factors that are recruited by USF-1, we performed tandem affinity purification (TAP) using USF-1 as bait, followed by mass spectrometric analysis (MS) (Supplementary Table 1). We identified JMJD1C, a member of JmjC-domain-containing HDMs, as a USF-1 interacting protein. The function of JMJD1C is not well studied but is reported to demethylate H3K9me2/me1[22]. By co-immunoprecipitation (co-IP), we verified the USF-1- JMJD1C interaction after transfection of Flag-tagged JMJD1C and HA-tagged USF-1into HEK293 cells (Fig. 1a). We next performed GST-pull-down using purified GST-USF-1a expressed in E. coli and pulled down in vitro translated JMJD1C, demonstrating the direct interaction. We also tested other TFs that are known to regulate lipogenesis, SREBP-1c, LXR, and ChREBP. None of these TFs directly interacted with JMJD1C, although SREBP-1c and LXR could make a complex with JMJD1C indirectly (Supplementary Fig. 1a, b). Overall, these results demonstrate the direct interaction of JMJD1C specifically with USF-1 for lipogenic gene transcription.

We next examined domains of JMJD1C and USF-1 for their interaction (Fig. 1b, left). Deletion of C-terminal basic Helix-Loop-Helix (bHLH) of USF-1 prevented the interaction with JMJD1C, indicating requirement of USF-1 bHLH domain for JMJD1C interaction (Fig. 1b, right). (Fig. 1c, left). Conversely, Co-IP clearly detected the USF-1 interaction with N-terminal quarter of JMJD1C (JMJD1C (1-619)), but not the other three regions of JMJD1C (Fig. 1c). We conclude that the bHLH domain of USF-1 directly interacts with N-terminal region of JMJD1C from aa 1 to 619.

We next tested the functional significance of the interaction between USF-1 and JMJD1C. Co-transfection of USF-1 with the −444-FAS promoter-luciferase construct increased luciferase activity by approximately sixfold. While transfection of JMJD1C alone did not have a significant effect, co-transfection of JMJD1C along with USF-1 resulted in an ~27-fold increase in FAS promoter activity (Fig. 1d, left and middle), demonstrating the synergistic activation of FAS promoter by USF-1 and JMJD1C. Treatment with Methylstat, an inhibitor of JmjC domain-containing demethylases, prevented activation of the FAS promoter, demonstrating the requirement of its demethylase activity (Fig. 1d, middle). Using the various JMJD1C domains we found that none of the four constructs, including the N-terminal JMJD1C (1–619) or the Jumonji catalytic domain could increase FAS-luciferase activity (Fig. 1d, right), suggesting that individually neither the USF-1 interacting N-terminal JMJD1C domain nor the JmjC domain are sufficient for FAS promoter activation.

Next, by chromatin immunoprecipitation (ChIP), we tested whether JMJD1C is recruited to the FAS promoter through USF-1. We detected JMJD1C bound to the FAS promoter region in insulin-treated HepG2 cells, but not in non-treated cells. JMJD1C was also enriched approximately five- to sixfold in the promoter regions of other lipogenic genes, such as ACC1 and SREBF1 (Fig. 1e, left). We detected Jmjd1c bound to the Fas promoter exclusively in the fed state (Fig. 1e, right). Jmjd1c was enriched also at Acc1 and Srebf1 promoters only in the fed state (Fig. 1e, right). In contrast, Jmjd1c was not detected in oxidative genes, such as Acox1. These results show that JMJD1C is recruited specifically to the lipogenic promoters in fed/insulin-treated conditions.

### JMJD1C activates lipogenic gene transcription in HepG2 cells.
Since JMJD1C is recruited specifically to activate lipogenic

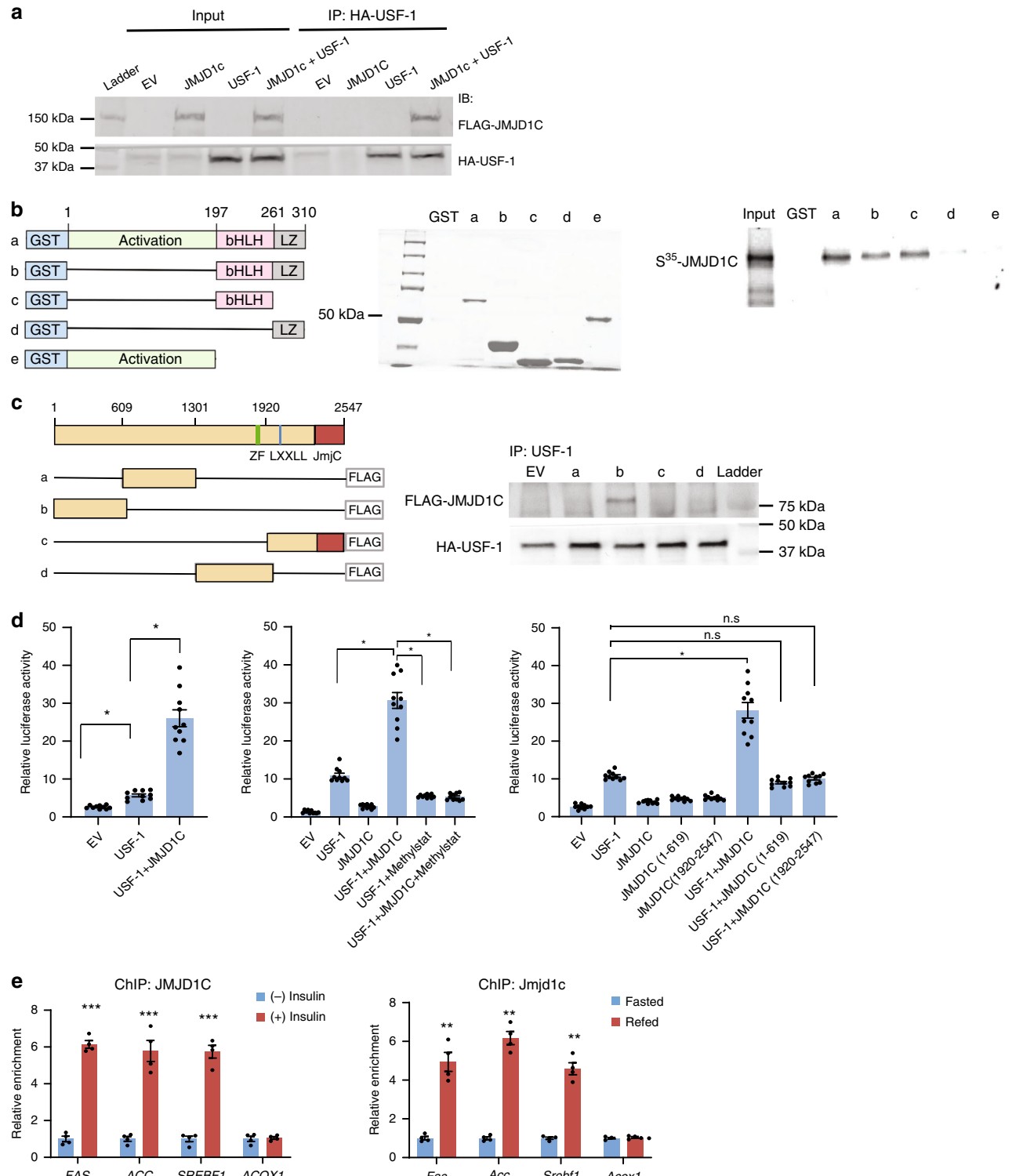

**Fig. 1 JMJD1C interaction with USF-1 for FAS promoter activation. a** IB of cell lysates of HEK293 cells co-transfected Flag-JMJD1C and HA-USF-1 with Flag antibody after IP with HA antibody (left). Immunoblotting of liver lysates from fasted and fed mice after IP with JMJD1C antibody (top right) and USF-1 antibody (bottom right). **b** Diagram of GST-USF-1 constructs (left). Coomassie Blue staining of SDS-PAGE of purified GST-USF-1 protein from bacterial lysates (middle). In vitro transcribed and translated $S^{35}$-methioine labeled JMJD1C was incubated with GST-USF-1 and subjected them to SDS-PAGE, followed by autoradiography (right). **c** Diagram of JMJD1C constructs (left). Co-IP of 293FT cells overexpressing Flag-tagged JMJD1C and USF-1. Immunoblotting with anti JMJD1C antibody after IP with USF-1 antibody (right). **d** FAS promoter activity in 293FT cells that we co-transfected with USF-1 with or without JMJD1C (left), with or without 10 μM Methylstat (Sigma), JMJD1C inhibitor (middle), and after overexpression of various deletions of JMJD1C (left). $n = 10$ wells of cells per group. Experiment was repeated three times. **e** ChIP for *FAS*, *ACC*, and *SREBPF1* promoters in HepG2 cells with or without insulin treatment (left, $n = 4$ per group) and ChIP for *Fas*, *Acc1*, and *Srebf1* promoters in liver from fasted or fed mice (right, $n = 3$ per group) after IP with JMJD1C antibody. Data are expressed as means ± SEM. **$P < 0.01$, ***$P < 0.001$, determined by two-tailed $t$-test. Source data are provided as a source data file.

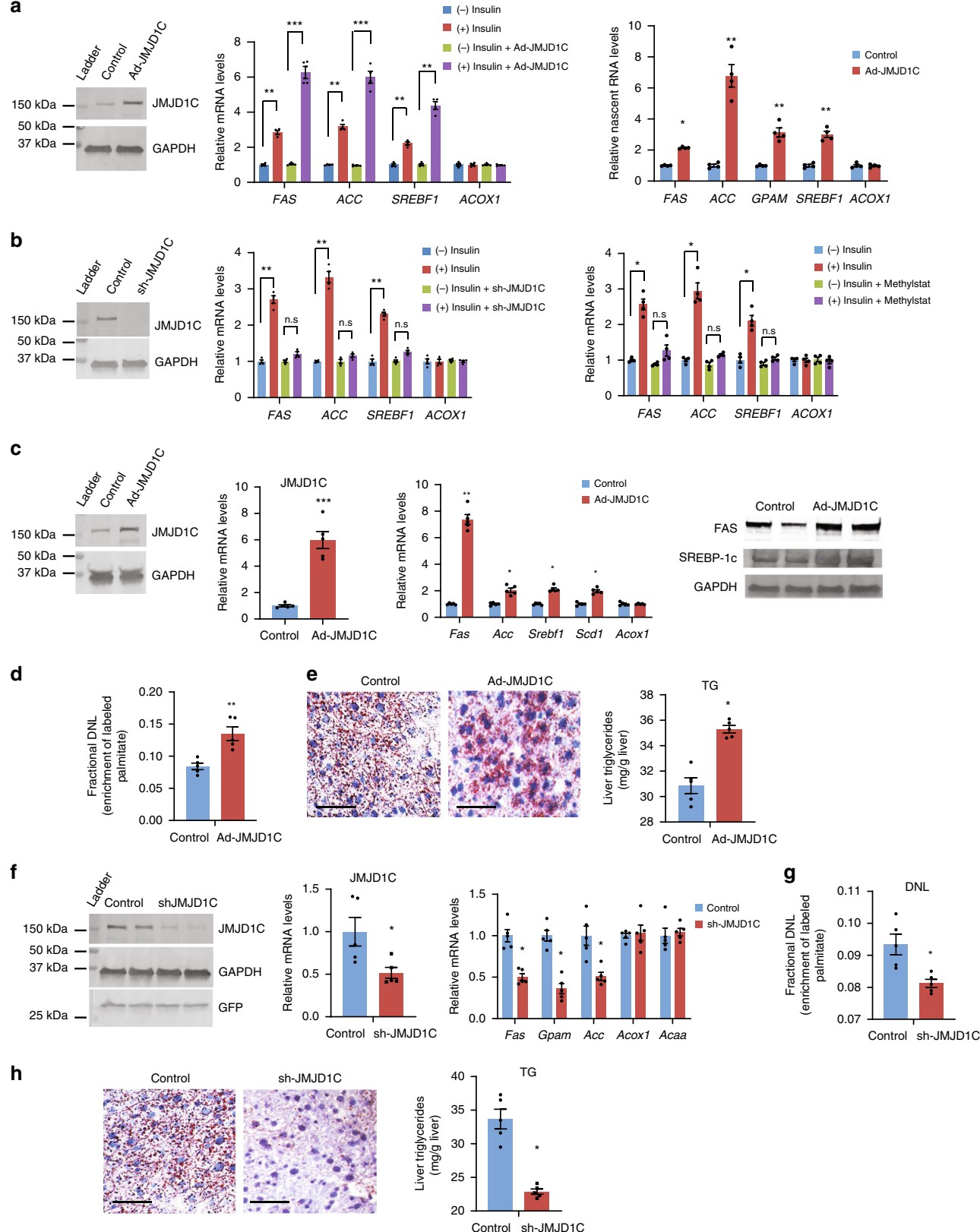

promoters, we examined lipogenic gene expression in response to insulin treatment by JMJD1C gain- and loss-of function. Adenoviral transduction of JMJD1C resulted in fourfold higher JMJD1C protein levels (Fig. 2a, left). This overexpression of JMJD1C did not affect lipogenic gene expression in HepG2 cells in the absence of insulin. *FAS*, *ACC*, *GPAM*, and *SREBF1*

messenger RNA (mRNA) levels were increased from four- to sevenfold upon insulin treatment in JMJD1C overexpressing cells, which were significantly higher than in control HepG2 cells that showed only two to threefold increase (Fig. 2a, middle). Similar changes in nascent RNA levels of these lipogenic genes were detected also (Fig. 2a, right). In contrast, mRNA and nascent

**Fig. 2 JMJD1C promotes lipogenesis. a** Immunoblotting of lysates from HepG2 cells infected with JMJD1C adenovirus (left). RT-qPCR for mRNA levels (middle) and nascent RNA levels (right) of lipogenic genes in HepG2 cells with or without insulin treatment. $n = 4$ wells of cells per group. Experiment was repeated three times. **b** Immunoblotting of lysates from HepG2 cells infected with sh-JMJD1C adenovirus (left). RT-qPCR for mRNA levels of lipogenic genes (middle) and mRNA levels of lipogenic genes in HepG2 cells with or without 10 μM Methylstat treatment (right). $n = 4$ wells of cells per group. **c** Immunoblotting for JMJD1C protein and RT-qPCR for mRNA levels (left) in livers of mice 10 days after tail-vein injection of $2 \times 10^8$ pfu JMJD1C adenovirus in PBS ($n = 5$). Mice were fasted for 16 h and then refed with HCD for 8 h before tissue harvesting. mRNA levels of lipogenic genes (middle). Immunoblotting of lysates from livers for Fas and Srebp-1c (right). **d** De novo lipogenesis, **e** Oil red O staining of liver tissue sections (left), and liver triglyceride levels (right) $n = 5$ per group. **f** Jmjd1c protein and mRNA levels (left two) in livers of mice 10 days after tail-vein injection of sh-JMJD1C adenovirus ($n = 5$). mRNA levels of lipogenic genes (right). **g** De novo lipogenesis, **h** Oil red O staining of liver tissue sections (left) and liver triglyceride levels (right). Data are expressed as means ± SEM. \*$P < 0.05$, \*\*$P < 0.01$, \*\*\*$P < 0.001$, determined by two-tailed $t$-test. Scale bar = 100 μm. Source data are provided as a source data file.

RNA levels of oxidative gene, *ACOX*, remained the same. We conclude that JMJD1C promotes lipogenic gene transcription in the presence of insulin. We next performed JMJD1C knockdown (KD) in HepG2 cells to test the requirement of JMJD1C for lipogenic gene activation in response to insulin. sh-JMJD1C adenoviral transduction led to a 50% decrease in JMJD1C protein levels (Fig. 2b, left). JMJD1C KD in the absence of insulin did not have any effect on lipogenic gene expression(Fig. 2b, middle). Moreover, expression levels remained low even in the presence of insulin in JMJD1C KD cells, while they were ~2.5- to 3.5-fold higher in control HepG2 cells in the presence of insulin (Fig. 2b, middle), demonstrating the requirement of JMJD1C for lipogenic gene activation in response to insulin. Treatment with Methylstat had a similar effect and prevented induction of lipogenic genes, even in insulin-treated cells (Fig. 2b, right).

**JMJD1C activates lipogenesis in response to feeding in mice.** Next, we examined in vivo significance of Jmjd1c in hepatic lipogenic gene activation. Administration of Jmjd1c adenovirus caused *Jmjd1c* mRNA levels to increase sixfold compared to the endogenous *Jmjd1c* levels in livers of mice (Fig. 2c, left). Upon feeding, *Fas* mRNA levels were increased sevenfold by JMJD1C overexpression. Similarly, other lipogenic genes, *Acc*, *Srebf1*, and *Scd1*, were also higher (Fig. 2c, middle). Protein levels of Fas and Srebp-1c were higher also (Fig. 2c, right). Hepatic DNL, determined by using heavy water-labeling followed by MS[23], increased ~1.5-fold upon Jmjd1c overexpression (Fig. 2d). Oil Red O (ORO) staining of liver sections showed higher lipid accumulation in Jmjd1c overexpressing livers (Fig. 2e, left). Liver TG content was also higher (Fig. 2e, right). We next administered Jmjd1c shRNA adenovirus, which decreased the Jmjd1c protein and mRNA levels livers by 60% (Fig. 2f, left). Hepatic mRNA levels of lipogenic genes, including *Fas*, *Gpam*, and *Acc1*, were significantly lower upon JMJD1C KD, while oxidative genes, such as *Acox1* and *Acaa*, did not change significantly (Fig. 2f, right). We detected DNL to decrease by 30% in Jmjd1c KD livers (Fig. 2g). Jmjd1c KD mice also had lower lipid staining in liver (Fig. 2h, left) and hepatic triglyceride levels showed a 25% decrease (Fig. 2h, right). These in vivo gain- and loss-of-function experiments demonstrate that Jmjd1c promotes lipogenic gene transcription and lipogenesis.

**JMJD1C ablation blunts hepatic lipogenesis in mice.** Jmjd1c is widely expressed and global JMJD1C KO mice are infertile[23]. To assess the liver-specific function of Jmjd1c, we generated liver-specific Jmjd1c knockout mice (designated JMJD1C-LKO) by crossing C57BL/6 JMJD1C-floxed mice with albumin-Cre mice (Fig. 3a, left). *Jmjd1c* mRNA level in livers of JMJD1C-LKO mice was decreased by 70%, but not in other tissues (Fig. 3a, middle). Jmjd1c protein was non-detectable in livers of JMJD1C-LKO mice (Fig. 3a, right). mRNA levels for lipogenic genes, including

*Fas*, *Gpam*, *Acc1*, and *Scd1* in livers of JMJD1C-LKO mice on chow diet, were ~50% lower compared to WT littermates (Fig. 3b, left). Fas and Srebp-1c protein levels were lower also (Fig. 3b, middle). We subjected JMJD1C-LKO mice to fasting/feeding cycle. Nascent RNA levels of lipogenic genes were drastically increased upon 6 h refeeding of high-carbohydrate (CHO) diet compared to fasting in WT mice. However, nascent RNA levels remained low in livers of JMJD1C-LKO mice even after feeding (Fig. 3b, right). ORO staining of livers showed lower lipid accumulation in fed JMJD1C-LKO mice (Fig. 3c, left). Liver TG content also was lower (Fig. 3c, right). Serum TG but not serum FFA levels were lower in JMJD1C-LKO mice than WT mice (Fig. 3d). Glycogen staining showed greater accumulation of glycogen in liver but not in muscle of JMJD1C-LKO mice (Fig. 3e and Supplementary Fig. 2D) These data support the concept that JMJD1C is required for induction of lipogenesis in livers of mice in response to feeding.

Next, we subjected JMJD1C-LKO mice to a high-CHO diet, a standard regimen for promotion of DNL. Similar to those on chow diet, mRNA and nascent RNA levels of lipogenic genes in livers of JMJD1C-LKO mice on high-CHO diet were significantly lower than in WT littermates (Fig. 3f). DNL rate was 33% lower in livers of JMJD1C-LKO mice (Fig. 3g). Liver TG levels also were lower in JMDJ1C-LKO mice (Fig. 3h–i), although liver TG levels in both WT and KO mice were higher than on chow diet (Supplementary Fig. 2c), demonstrating that Jmjd1c ablation protected mice from hepatosteatosis from high-CHO diet regimen. Serum TG levels of JMJD1C-LKO mice compared to WT littermates were lower (Fig. 3j). In contrast, glycogen levels upon refeeding were higher in the JMJD1C-LKO livers (Fig. 3k). Neither WT nor JMJD1C-LKO mice on high-CHO diet had higher white adipose tissue (WAT) mass or adiposity compared to chow fed mice (Fig. 3l). However, GTT and ITT showed IR in WT mice. In contrast, GTT and ITT showed significantly lower glucose levels in JMJD1C-LKO mice, compared to the WT (Fig. 3m) and were similar to when on chow diet (Supplementary Fig. 2a, b), demonstrating that Jmjd1c ablation prevented glucose intolerance and IR from high-CHO feeding.

We also examined the potential involvement of Jmjd1c in lipogenic gene activation by Srebp-1c and Lxr[24,25]. We overexpressed Srebp-1c by adenoviral transduction or administered Lxr agonist, T0901317. As expected, WT mice showed a 2–3-fold increase in lipogenic gene expression by either of these treatments (Supplementary Fig. 3b, c). This lipogenic gene induction was partially blunted in JMJD1C-LKO mice, demonstrating involvement of Jmjd1c in Srebp-1c and Lxr function, probably via their indirect interaction (Supplementary Fig. 1) and/or these TFs may require chromatin accessibility brought by Jmjd1c at the lipogenic promoters (see below).

We next subjected JMJD1C-LKO mice to high-fat diet (HFD) to test whether Jmjd1c affects diet-induced obesity. Both WT and JMJD1C-LKO mice on HFD showed a typical obesity phenotype

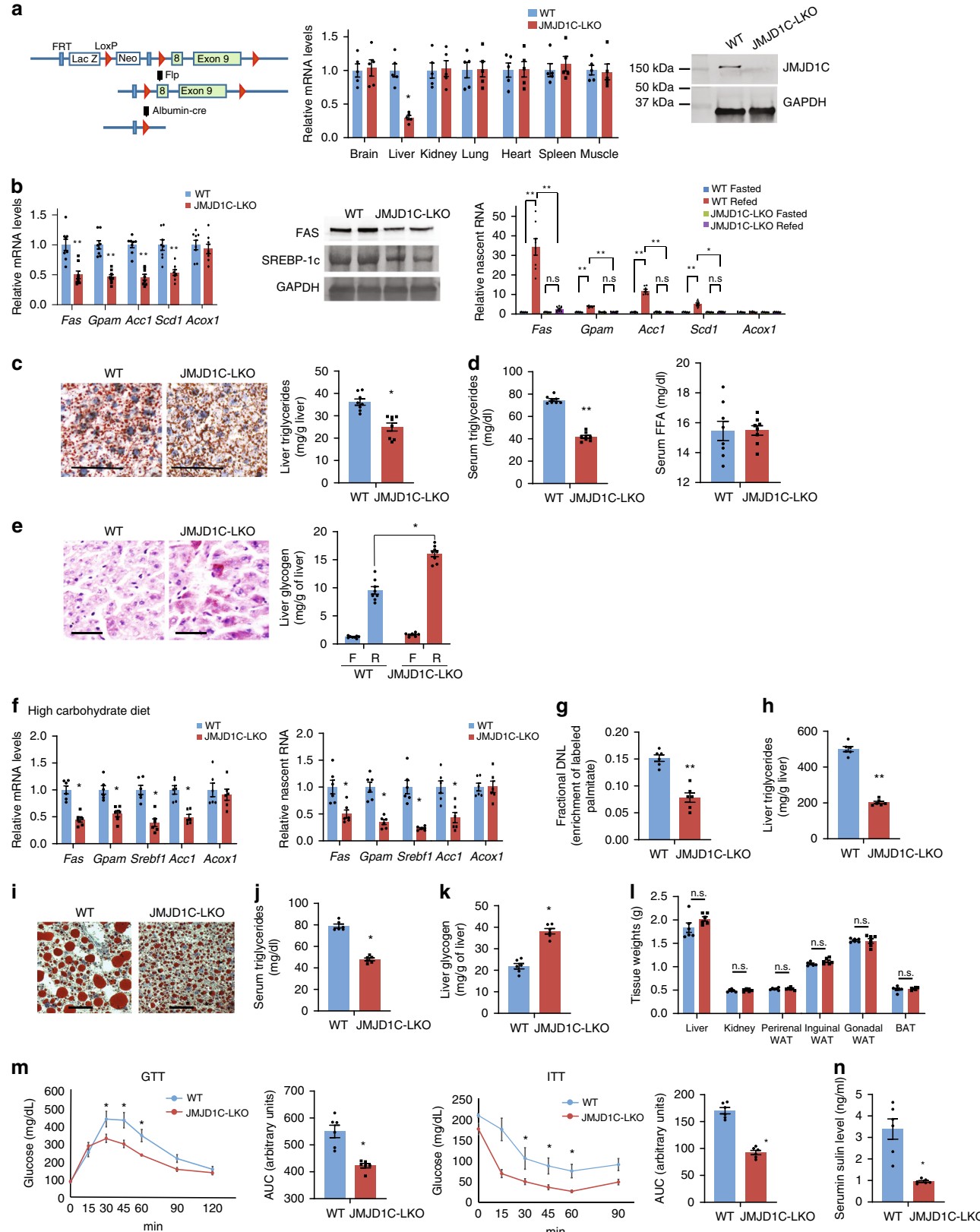

with higher WAT mass (Supplementary Fig. 2e). WT mice showed liver TG accumulation and development of glucose intolerance (Supplementary Fig. 2f, h). In contrast, JMJD1C-LKO mice on HFD remained insulin-sensitive, protected from diet-induced obesity-related IR (Supplementary Fig. 2h), even when these mice still manifested higher WAT mass. These results demonstrate that Jmjd1c ablation can improve IR even in the presence of obesity caused by HFD.

**JMJD1C demethylates H3K9 at lipogenic promoters.** Although JMJD1C is reported to demethylate H3K9[22,26], little is known

**Fig. 3 Lipogenesis is impaired in liver-specific JMJD1C knockout mice. a** Schematics of conditional knockout allele of JMJD1C (left). RT-qPCR for *Jmjd1c* mRNA levels in tissues, *n* = 5 mice (middle) and immunoblotting of liver tissue lysates with JMJD1C antibody (right). **b** RT-qPCR for mRNA (left), immunoblotting of liver tissue lysates with Fas and Srebp-1c antibodies (middle), and nascent RNA (right). **c** Oil red O staining of liver tissue sections and liver triglyceride level. **d** Serum triglyceride and FFA levels in 12-week-old mice on chow diet. **e** Periodic acid–Schiff (PAS) staining for glycogen of liver tissues and glycogen levels in liver. **a–e** *n* = 8 mice per group. **f–n** JMJD1C-LKO mice and their WT littermates were fed a High CHO (70 kcal% carb) for 4 months after weaning, *n* = 6 mice per group. **f** mRNA (left) and nascent RNA (right), **g** de novo lipogenesis, **h** liver triglycerides, **i** Oil red O staining of liver tissue sections, **j** serum triglyceride levels, **k** liver glycogen levels, **l** and tissue weights. **m** GTT and ITT for high-CHO diet, and **n** serum insulin level. Data are expressed as means ± SEM. *$P < 0.05$, **$P < 0.01$, determined by two-tailed *t*-test. Scale bar = 100 μm. Source data are provided as a source data file.

regarding the change in H3K9 methylation at lipogenic promoters upon insulin/feeding. By ChIP–quantitative PCR (qPCR), we detected ~60% lower H3K9me3 enrichment at the promoters of lipogenic genes, such as *FAS*, *ACC*, and *SREBF1*, in insulin-treated HepG2 cells (Fig. 4a, top). H3K9me2 enrichment was also 30–55% lower, while H3 level remained constant (Fig. 4a). Conversely, H3K9me1 enrichment increased twofold upon insulin treatment. We detected approximately sevenfold greater enrichment of H3K4me3 upon insulin treatment (Fig. 4b). Changes in H3K9 methylation were specific to lipogenic genes, since we did not detect changes in promoter regions of oxidative genes, such as *ACOX*. ChIP experiments using livers from fasted and fed mice showed lower enrichment of H3K9me3 and H3K9me2, and greater enrichment of H3K9me1 at lipogenic promoters in livers of fed mice (Fig. 4a). Enrichment of H3K4me3 was higher in refed mice (Fig. 4b).

We next performed in vitro demethylation assays using individual H3K9me3, H3K9me2, or H3K9me1 peptides to test JMDJ1C activity. We detected nearly complete demethylation of the H3K9me2, ~50% demethylation of H3K9me1, and virtually no demethylation of the H3K9me3 (Fig. 4c). To define the role of JMJD1C in H3K9 demethylation in the activation of lipogenic genes, we next compared H3K9 methylation status at the lipogenic promoter regions in livers of JMJD1C-LKO and WT mice. We detected greater enrichment of H3K9me3, H3K9me2, and H3K9me1 at the lipogenic promoters, including *Fas* and *Gpam*, but not in *Acox1*, in livers of JMJD1C-LKO, compared to WT mice in the fed state (Fig. 4d, left and middle). (Fig. 4d, right). Interestingly, though JMJD1C had no H3K9me3 demethylase activity, we still detected accumulation of H3K9me3 in JMJD1C-LKO mice. It is possible that a yet to be identified HMT for H3K9 is not regulated by fasting/feeding and thus accumulation of H3K9me2 caused H3K9me3 accumulation. Overall, our results support the concept that JMJD1C demethylates H3K9me2 for activation of lipogenic genes upon feeding/insulin treatment.

**Genome-wide analyses of JMJD1C function in livers of mice**. With the known function of JMJD1C as an H3K9 HDM, JMJD1C is predicted to play a role in regulating the transcriptional landscape. We therefore assessed the chromatin landscape by Assay for Transposase-Accessible Chromatin (ATAC-seq). ATAC-seq showed concentrated open chromatin regions near the transcription start sites (TSS), in both WT fasted and refed samples, as well as refed JMJD1C-LKO (Fig. 5a). Approximately 38% of peaks to be at the TSS, 25% in the intragenic regions, while 37% were spread out throughout intergenic regions (Supplementary Fig. 5a). Gene ontology identified lipogenesis exclusively in the refed sample, while those genes involved in processes of gluconeogenesis appeared exclusively in fasted samples (Fig. 5b, top). RNA processing and translation, along with various lipid biosynthetic processes were detected exclusively in WT, but not JMJD1C-LKO (Fig. 5b, bottom). Genes involved in chromatin organization such as *Hdac*, *Arid*, *Jarid*, *Hmt*, *Set*-domain containing, and *Smarca* families that are known to alter chromatin

structure to repress transcription[27–30], were detected in open chromatin in both WT and JMJD1C-LKO (Fig. 5b, bottom). Motif analysis detected 55 known transcription factor motifs that were significantly enriched in the WT but not in the JMJD1C-LKO (Supplementary Table 2). Interestingly the motif for Usf-1 was among these (Supplementary Table 2), suggesting that Jmjd1c ablation impacted Usf-1 function other than in lipogenic gene regulation. With our primary interest in assessing JMJD1C's role in the regulation of lipogenesis, we compared peaks for a subset of lipid biosynthetic genes. Indeed, we found that lipogenic promoters were more open upon feeding than fasting (Fig. 5c). Conversely, promoter regions of gluconeogenic genes were more open during fasting than refeeding, while the promoter region of *Jmjd1c* was not different between fasting/feeding, consistent with our finding that *Jmjd1c* expression was not regulated by fasting/feeding (Fig. 5c and Supplementary Fig. 5). Comparing WT peaks to that of the JMJD1C-LKO showed much smaller peaks and no apparent change as a result of feeding (Fig. 5c) at lipogenic promoters.

We performed ChIP-seq that detected greater enrichment of H3K9me2 at the TSS in fasted and refed samples, however H3K9me2 enrichment was detected also in the proximal promoter regions, as well as within gene bodies (Fig. 5d). Approximately 46% of peaks were found near the TSS, 50% were found in intragenic regions, and 4% were found in intergenic regions (Supplementary Fig. 5b). H3K9me2 enrichment upon refeeding was detected at genes from processes known to be suppressed, such as those for gluconeogenesis and MAPK signaling (Fig. 5e, left). Similarly, H3K9me2 enrichment in fasted livers was detected at those genes repressed during fasting, such as those involved in lipogenesis, as well as insulin and TOR signaling (Fig. 5e, left). Comparing ATAC-seq and H3K9me2 ChIP-seq of refed livers, showed distinct gene profiles (Fig. 5e, right). Open chromatin regions were found in lipogenic promoters and were not enriched in H3K9me2, whereas gluconeogenic or fatty acid oxidative genes were found to be H3K9me2 enriched, but not detected among open chromatin regions in the refed livers. Among the genes exclusive to either ATAC-seq or H3K9me2 ChIP-seq datasets, Chromatin Organization again was the most enriched process. We then examined peaks for representative lipogenic (*Fas*) and oxidative (*Acot1*) promoters and compared them to those we detected in ATAC-seq. ATAC-seq showed open chromatin regions for *Fas* only in the refed, while that of *Acot1* was only open in the fasted (Fig. 5f). However, H3K9me2 ChIP-seq from refed livers detected peaks for *Acot1*, but not for *Fas* (Fig. 5f), showing repression of *Acot1* but not *Fas* during refeeding.

We next performed RNA-seq to determine the effect of *Jmjd1c* depletion in the liver. Protein synthesis was detected to be the most affected process downregulated in JMJD1C-LKO mice, with various initiation/elongation factors, ribosomal proteins, and transfer RNA (tRNA) synthetases being downregulated (Fig. 5g, left). Chromatin modification, regulation of cell cycle, lipogenesis, and DNA maintenance were also downregulated in JMJD1C-

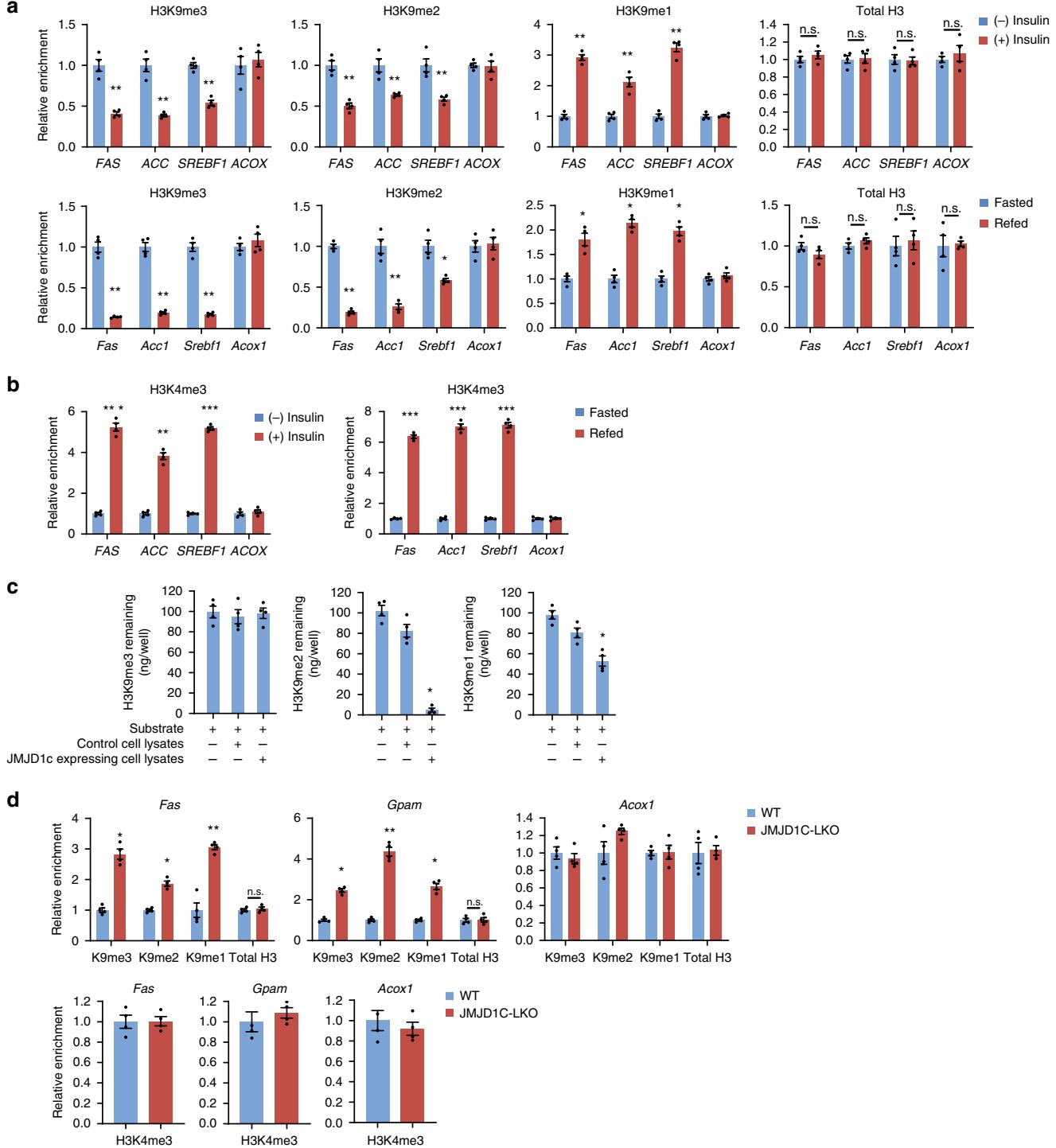

**Fig. 4 Enrichment of H3K9me2 is decreased on lipogenic promoters by feeding/insulin. a** ChIP for enrichment of H3K9me3, H3K9me2, and H3K9me1 on *FAS*, *ACC*, and *SREBF1* promoters in HepG2 cells with or without insulin treatment (top) *n* = 4 wells of cells per group. ChIP for enrichment of H3K9me3, H3K9me2, and H3K9me1 on lipogenic gene promoters in livers from fasted and fed mice (bottom) *n* = 4 per group. **b** ChIP for enrichment of H3K4me3 on lipogenic promoters upon insulin/feeding, *n* = 4 per group. **c** In vitro demethylation assays using individual H3K9me3, H3K9me2, or H3K9me1 peptides, along with control or JMJD1C expressing HepG2 cell lysates, *n* = 4 dishes of cells per group. **d** ChIP for enrichment of H3K9me3, H3K9me2, and H3K9me1 on *Fas* and *Gpam* promoters in livers from JMJD1C-LKO and their WT littermates on chow diet in the fed state. *n* = 8 per group. Data are expressed as means ± SEM. *$P < 0.05$, **$P < 0.01$, determined by two-tailed *t*-test. Source data are provided as a source data file.

LKO mice (Fig. 5g, left). Genes in various metabolic processes were also detected to be upregulated in the JMJD1C-LKO mice (Fig. 5g, left). The most significantly affected was glycolysis, with phosphofructokinase, aldolase, and phosphoglycerate kinase, among others, being upregulated in the JMJD1C-LKO livers.

We next compared genes downregulated in the JMJD1C-LKO mice to those detected in refed WT mice by ATAC-seq. We reasoned that if these genes were directly downregulated due to depletion of *Jmjd1c*, they should coincide with genes detected in the refed WT by ATAC-seq. We detected similar processes

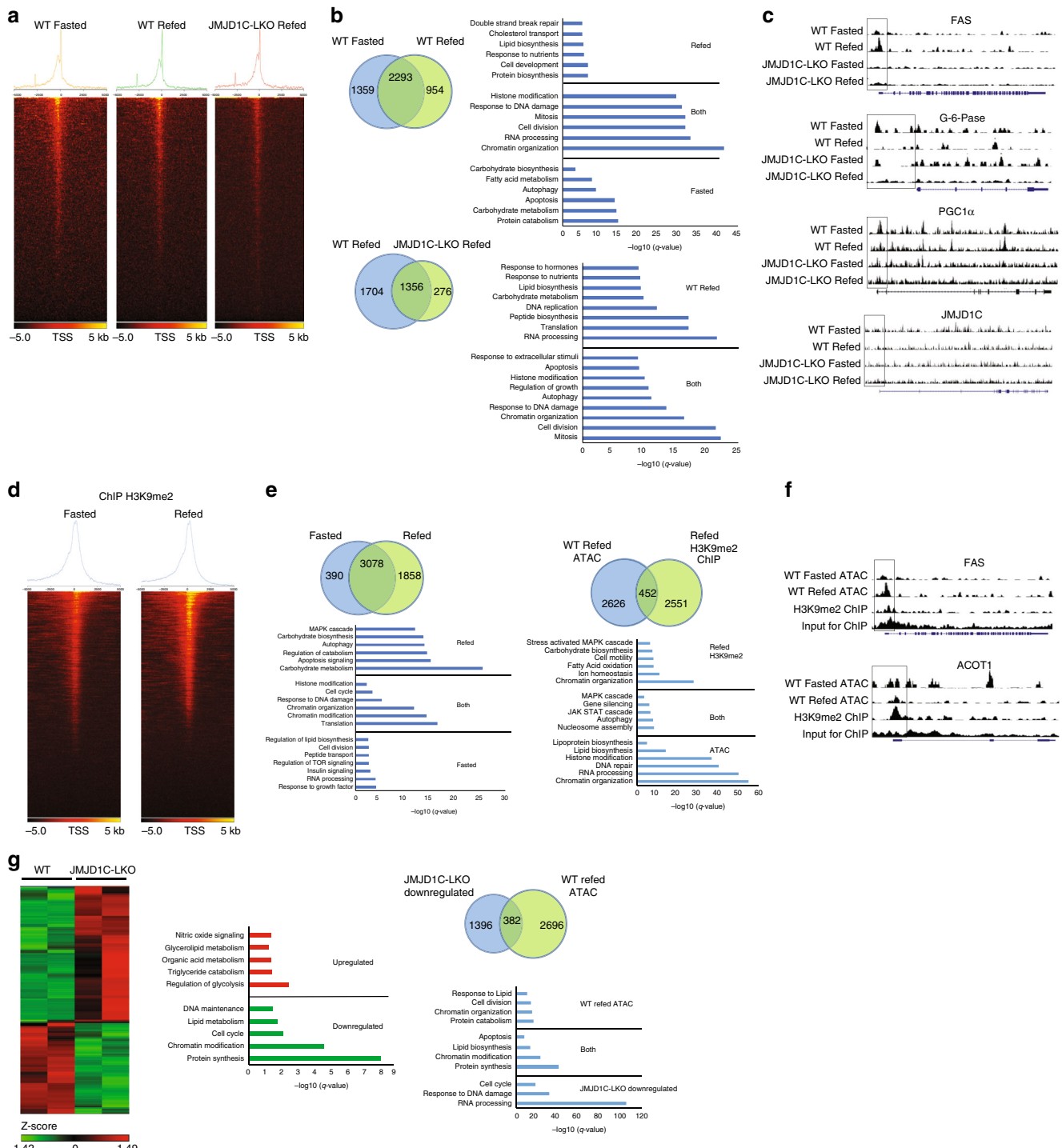

**Fig. 5 Genome-wide analyses of JMJD1C function in liver. a–c** ATAC-seq using livers from WT and JMJD1C-LKO mice that were fasted or refed. $n = 2$ per group. **a** Heatmaps showing open chromatin regions focused at the transcription start site (TSS). Color scale shows peaks detected, black is equivalent to zero, yellow is equivalent to 500. **b** Venn diagrams showing number of unique or shared genes and charts for representative top gene ontology (GO) terms. **c** UCSC genome browser screenshot of representative peaks at a subset of promoter regions. **d–f** H3K9me2 ChIP-seq using livers from WT refed mice, $n = 2$ per group. **d** Heatmap showing H3K9me2 enrichment at TSS (color scale shows peaks detected, black is equivalent to zero, yellow is equivalent to 500) and **e** representative top GO terms for genes detected to be H3K9me2 enriched differentially upon fasting or refeeding. Venn diagrams showing number of unique or shared genes between ATAC- and ChIP-seq datasets and charts for representative top gene ontology (GO) terms (right). **f** UCSC genome browser screenshot of representative peaks at a subset of promoter regions. **g** RNA-seq using livers from refed WT and JMJD1C-LKO mice, $n = 2$ pooled RNA samples per group. Heatmap showing changes in gene expression between WT and JMJD1C-LKO mice (left), Color scale shows changes in gene expression as determined by $Z$-score, green is –1.42 and red is 1.49. Representative top GO terms of upregulated and downregulated genes identified by differential expression analysis (middle). Venn diagrams showing number of unique or shared genes between ATAC- and RNA-seq datasets and charts for representative top gene ontology (GO) terms (right).

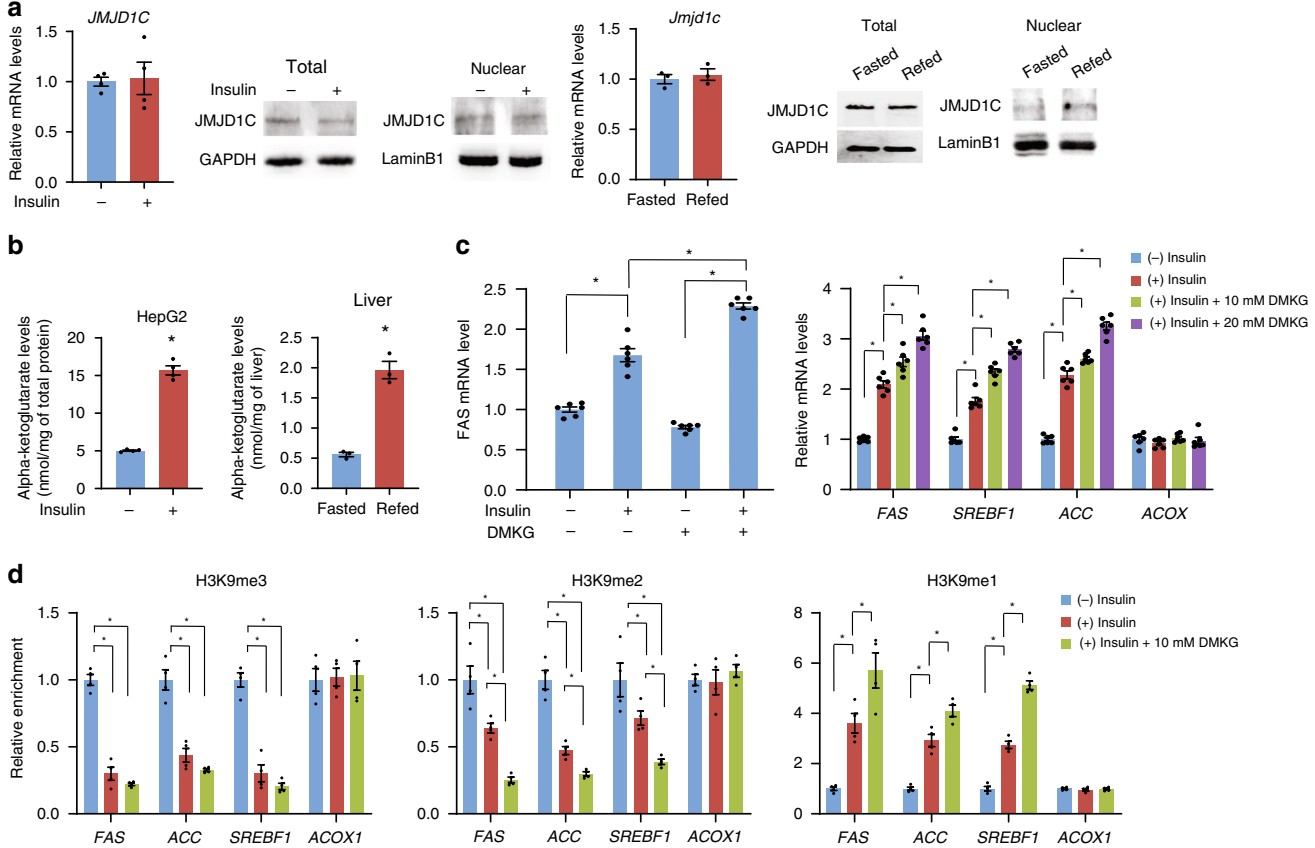

**Fig. 6 α-KG level increases upon feeding/insulin treatment in liver for JMJD1C activity. a** RT-qPCR for *JMJD1C* mRNA level and immunoblotting with anti- JMJD1C antibody for JMJD1C protein in whole-cell lysates and in nuclear extracts from HepG2 cells with or without insulin treatment (left three), *n* = 4 wells of cells per group. RT-qPCR for *JMJD1C* mRNA level and immunoblotting for JMJD1C protein in whole-tissue lysates and in nuclear extraction of livers from fasted or fed mice (right three), *n* = 3 per group. **b** intracellular levels of α-ketoglutarate levels in HepG2 cells with or without 100 nM insulin treatment for 30 min, *n* = 4 wells of cells per group (left) and in livers from fasted and refed mice, *n* = 3 per group (right). **c** RT-qPCR for *FAS* mRNA levels in HepG2 cells treated with or without 10 mM DMKG treatment for 8 h (left). RT-qPCR for mRNA levels of lipogenic genes with DMKG at indicated concentration (right), *n* = 6 wells of cells per group. **d** ChIP assay for enrichment of H3K9me3, H3K9me2, and H3K9me1 on lipogenic gene promoters in HepG2 cells with insulin and/or 10 mM DMKG treatment, *n* = 4 wells of cells per group. Data are expressed as means ± SEM. *P < 0.05, determined by two-tailed *t*-test. Source data are provided as a source data file.

between the two datasets, including protein synthesis, chromatin modification, lipogenesis, and apoptosis (Fig. 5g, right). Genes involved in protein synthesis detected in both datasets included initiation/elongation factors, ribosomal proteins, and tRNA synthetases. The chromatin modifying genes found in both ATAC and RNA-seq datasets were composed of *Arid*, *Jarid*, *Hdac*, and *Smarca* family members, which are known to function primarily in transcriptional repression. From these comparisons, it is clear that JMJD1C plays a critical role in the activation of transcription of lipogenic genes. Moreover, JMJD1C also appears to play an important role in regulating the expression of other histone modifiers and chromatin remodelers.

**JMJD1C is phosphorylated at T505 by mTOR.** We next asked how JMJD1C activates lipogenic gene transcription in response to feeding/insulin treatment. *JMJD1C* mRNA and protein levels remained the same in HepG2 cells, whether treated with insulin or not (Fig. 6a, left), and in livers of fasted or fed mice (Fig. 6a, right). Therefore, changes in JMJD1C levels cannot explain JMJD1C-mediated action on lipogenesis in response to insulin/ feeding. Additionally, we did not detect any changes in nuclear levels of JMJD1C, eliminating translocation as an underlying mechanism. Since α-ketoglutarate (αKG) serves as an obligatory co-factor for the Jumonji HDMs, we examined if αKG levels are

altered in response to insulin/feeding, and detected a threefold increase in intracellular αKG in insulin-treated HepG2 cells (Fig. 6b, left). Similarly, αKG levels were higher in livers of refed mice (Fig. 6b, right). Moreover, treatment of HepG2 cells with a cell-permeable analog of αKG, dimethyl-ketoglutarate (DMKG), further increased *FAS* and other lipogenic gene expression in a dose-dependent manner (Fig. 6c). We also examined H3K9 methylation status of lipogenic promoters by ChIP following insulin/DMKG treatment. Insulin-treated cells had lower enrichment of H3K9me3 and H3K9me2, and greater enrichment of H3K9me1(Fig. 6d). Cells to which insulin and DMKG were added had even lower enrichment of H3K9me2, and greater accumulation of H3K9me1, at lipogenic promoters compared to the insulin-treated cells (Fig. 6d).

To examine if changes in αKG levels can fully explain the JMJD1C-mediated activation of lipogenic genes in response to insulin/feeding treatment, we next assayed for the demethylase activity of JMJD1C purified from HepG2 cells in the excess presence of all of the required co-factors and substrates, including αKG. Notably, we detected significantly higher demethylase activity of JMJD1C purified from insulin-treated HepG2 cells (Fig. 7a). Similarly, we detected higher demethylase activity with the immuno-purified JMJD1C from livers of fed mice (Fig. 7a). Higher demethylase activity of JMJD1C from insulin/feeding

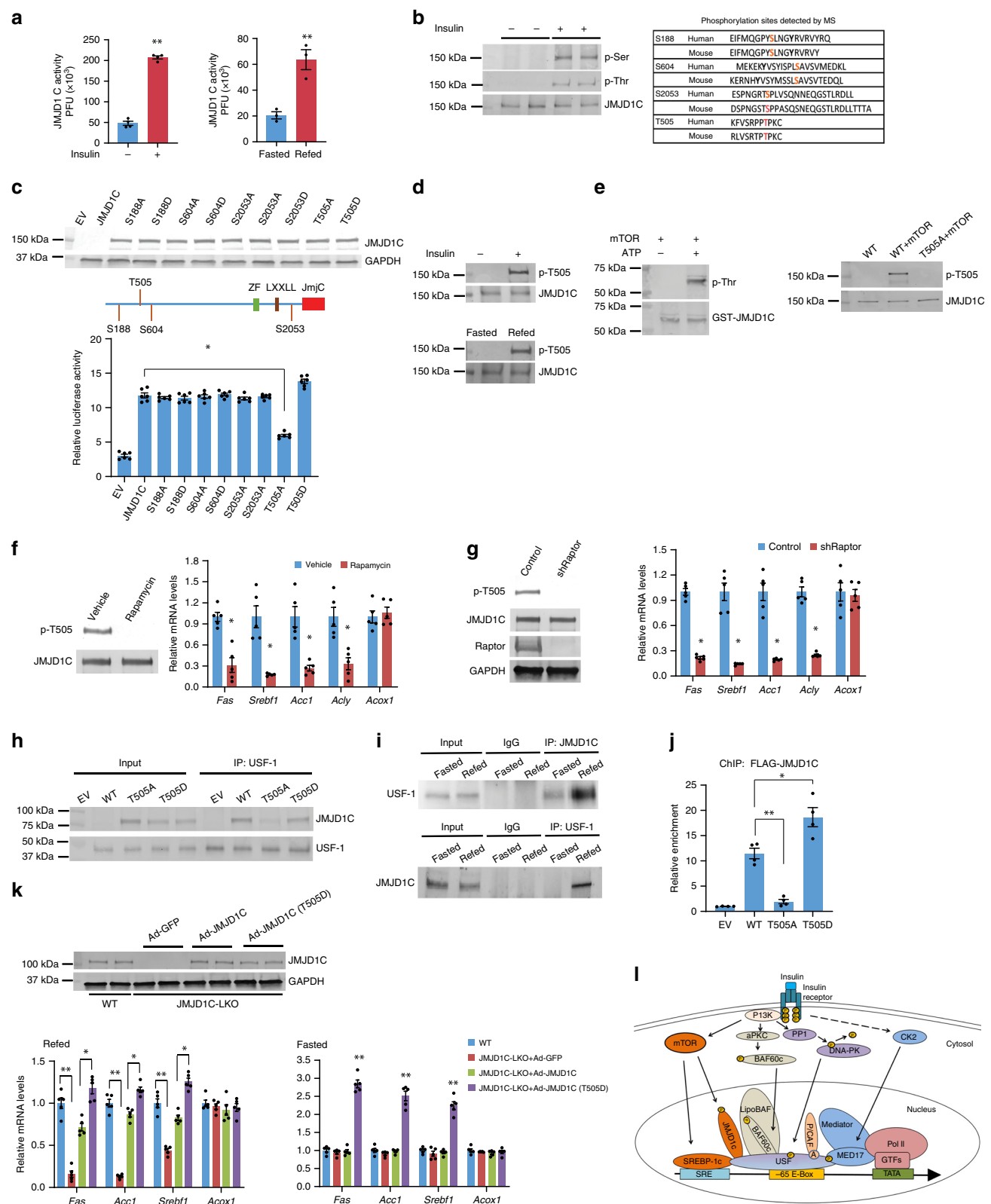

treatment, suggests that JMJD1C could be posttranslationally modified in response to insulin/feeding.

We next examined the possibility of posttranslational modification of JMJD1C. Immunoblotting using pan-p-Ser antibody detected signals in both insulin-treated and non-treated cells, although the signal was somewhat stronger in insulin-treated cells (Fig. 7b, left). In contrast, immunoblotting with pan-p-Thr antibody showed phosphorylation only in insulin-treated cells, but not in non-treated cells (Fig. 7b, left). We then subjected immuno-purified JMJD1C, from insulin-treated, JMJD1C-HA transduced HepG2 cells to MS analysis and identified four phosphor-Ser-containing peptides with S188, S604, S607, and S2053, as well as one phospho-Thr-containing peptide with T505 (Fig. 7b, right). We generated site-specific mutation constructs

**Fig. 7 JMJD1C is phosphorylated at T505 by mTOR in response to insulin/feeding. a** JMJD1C demethylase activity in HepG2 cells with or without insulin treatment, $n = 4$ wells of cells per group (left) and in livers from fasted and fed mice, $n = 3$ per group (right). **b** Immunoblotting with phospho-serine (p-Ser) and threonine (p-Thr) antibodies using lysates from HepG2 cells with or without Insulin treatment (left). Phosphorylated peptides detected by mass spec analysis (right). **c** Diagram for phosphorylation sites detected by mass spec analysis (top). FAS promoter activity in HEK293 that were transfected with JMJD1C serine or threonine mutants, $n = 6$ wells of cells per group (bottom). **d** IB with JMJD1C phospo-T505-specific antibody. JMJD1C T505 phosphorylation in HepG2 cells upon insulin treatment (top) in livers of mice upon feeding (bottom). **e** Purified GST-JMJD1C from overexpression in *E.coli* was incubated with mTOR complex with or without ATP. Immunoblotting with anti-phospho-threonine antibody after SDS-PAGE separation (left). IB with phospho-T505 antibody after co-transfection with JMJD1C WT or T505A mutant with mTOR complex, after IP with JMJD1C antibody (right). **f** Treatment of mice with vehicle or rapamycin (10 mg/kg), IB for phosphor-T505 and total Jmjd1c (left) and lipogenic gene mRNA levels in liver (right) $n = 5$. **g** Knockdown of Raptor in mouse liver, IB for phospho-T505, total Jmjd1c, Raptor, and Gapdh (left), and mRNA levels of lipogenic genes (right), $n = 5$. **h** IB following co-IP of JMJD1C mutants with USF-1. **i** IB of Usf-1 following IP of Jmjd1c (top) and IB of Jmjd1c following IP of Usf-1 (bottom) from mouse fasted and refed mouse liver. **j** ChIP–qPCR using FAS promoter luc construct to assess enrichment of JMJD1C mutants upon insulin treatment, $n = 4$ wells of cells per group. **k** IB for Jmjd1c from livers of WT and JMJD1C-LKO mice injected with designated virus (top). RT-qPCR for lipogenic gene mRNA levels in livers from WT and JMJD1C-LKO mice 10 days after tail-vein injection of JMJD1C or its mutant T505D adenovirus in refed (left) and fasted (right) conditions, $n = 5$ per group. **l** Representative schematic of insulin signaling pathways converging on USF-1 for recruitment of JMJD1C and other co-factors for the activation of transcription of lipogenic genes. Solid lines show established pathways, dashed lines show undefined pathways. **a–k** Data are expressed as means ± SEM. *denotes $p < 0.05$, determined by two-tailed $t$-test. Source data are provided as a source data file.

replacing Ser or Thr to non-phosphorylatable Ala mutants, as well as phosphorylation-mimicking Asp mutants for S188, S604, S2053, and T505 for FAS-Luc assay. None of the Ser mutants changed the FAS promoter activity (Fig. 7c). However, the T505A mutant showed 40% lower FAS promoter activity compared to the WT JMJD1C (Fig. 7c). The phosphorylation-mimicking T505D mutant showed higher FAS activity, although the magnitude of increase was small, probably because the cells were maintained in the presence of insulin causing phosphorylation of the WT JMJD1C (Fig. 7c). These results support that JMJD1C T505 phosphorylation plays a key role in the activation of the FAS promoter by JMJD1C. We then generated an antibody specific to phosphorylated T505 (p-T505) to examine insulin/feeding-induced phosphorylation of JMJD1C. Indeed, we detected phosphorylation of T505 of the endogenous JMJD1C exclusively in insulin-treated HepG2 cells and not in non-treated cells (Fig. 7d, top). We also detected T505 phosphorylation of the endogenous JMJD1C from livers of fed, but not fasted, mice (Fig. 7d, bottom). A review of phospho-protein databases predicted that mTOR could potentially phosphorylate the T505 of JMJD1C. By performing in vitro phosphorylation assay using mTORC1 (including mTOR, RAPTOR, and Rheb), we detected phosphorylation of T505 of JMJD1C (Fig. 7e, left). Furthermore, we detected phosphorylation of WT JMJD1C but not T505A mutant when we co-transfected JMJD1C constructs along with the mTORC1 in 293FT cells (Fig. 7e, right). Moreover, rapamycin treatment completely blocked phosphorylation of Jmjd1c at T505, and decreased lipogenic gene expression by 60–70% (Fig. 7f). Administration of adenoviral shRNA for Raptor blocked Jmjd1c T505 phosphorylation (Fig. 7g) and lipogenic gene expression was 80% lower in Raptor KD livers. These results indicate that T505 of JMJD1C is phosphorylated by mTORC1 upon insulin/feeding.

We next tested whether the JMJD1C interaction with USF-1 is affected by JMJD1C phosphorylation. Indeed, Co-IP experiments detected interaction of USF-1 with WT and the T505D mutant but not with T505A mutant, showing that T505 phosphorylation enhances interaction with USF-1 (Fig. 7h). Co-IP showed a strong interaction between endogenous USF-1 and JMJD1C only in livers from fed but not fasted mice (Fig. 7i). We also detected an interaction between JMJD1C and LipoBAF complex member BAF60c, which was exclusively detected in refed livers (Supplementary Fig. 6). We next examined the effect of JMJD1C phosphorylation on recruitment to the FAS promoter. (Fig. 7j). By ChIP, we detected ~12-fold enrichment of WT JMJD1C at the FAS promoter, and ~17-fold enrichment of the T505D JMJD1C

mutant. Enrichment of the T505A JMJD1C mutant was considerably lower than both WT and T505D JMJD1C (Fig. 7j). These results indicate that T505 phosphorylation enhances JMJD1C-USF-1 interaction for recruitment to the FAS, and likely other lipogenic promoters.

Finally, to evaluate the effect of JMJD1C T505 phosphorylation in vivo, we injected control GFP, WT JMJD1C, or JMJD1C T505D mutant to JMJD1C-LKO mice. Protein levels of various JMJD1C forms were similar (Fig. 7k). Compared to WT control, hepatic levels of lipogenic genes, including *Fas*, *Acc1* and *Srebf1*, were lower by 60–90% in JMJD1C-LKO mice (Fig. 7k, left). Administration of WT JMJD1C restored the gene expression up to 75–80% of WT levels. More importantly, administration of JMJD1C containing T505D mutation increased lipogenic gene expression 20-30% higher than WT JMJD1C (Fig. 7k, left). Expression of oxidative genes, such as *Acox1*, was not affected, revealing its specificity for lipogenic genes. We next examined the effects of JMJD1C mutants in mice during fasting. WT mice, along with JMDJ1C-LKO mice injected with GFP or Ad-JMJD1C maintained low mRNA levels of lipogenic genes, whereas JMJD1C-LKO mice injected with Ad-JMJD1C (T505D) had 2–3-fold higher mRNA levels of lipogenic genes even during fasting when lipogenic genes were suppressed (Fig. 7, bottom right), evidence of the critical function of T505 phosphorylation for lipogenic gene activation. Overall, we conclude that mTOR catalyzed JMJD1C T505 phosphorylation mediates activation of lipogenic gene transcription upon insulin/feeding.

## Discussion
Coordinate transcriptional regulation of lipogenic genes is one the most exquisitely regulated processes in mammalian metabolism. We previously reported the critical role of USF-1, which is phosphorylated via a specific insulin signaling pathway involving DNA-PK to recruit other factors, including P/CAF, and BAF60c, to various lipogenic genes for transcriptional activation[6,7]. However, these factors require accessible chromatin landscape from proper histone modifications. We now determine that JMJD1C is recruited by USF-1 to various lipogenic genes for H3K9 demethylation to enhance chromatin accessibility in the fed state. Moreover, we demonstrate that this is via mTORC phosphorylation of JMJD1C at T505 to allow direct interaction with USF-1 and via an increase in intracellular αKG levels, upon feeding/insulin.

Histone modification, a major mode of epigenetic regulation, is influenced by nutrients to alter chromatin accessibility[12,16]. One

such histone modification is H3K9 methylation[14,31]. H3K9me2 and H3K9me3 are reported to be enriched in the TSS of silenced genes and are correlated with transcriptional repression. H3K9me1 could also be found in the promoter regions of active or silent genes, although its role remains controversial[32–36]. Numerous H3K9 demethylases have been identified and most of them belong to the Jumonji family with 30 members. Here, we report that JMJD1C, one of Jumonji HDM family members, functions specifically for lipogenic gene activation in response to feeding/insulin. By direct interaction, USF-1 recruits JMJD1C to lipogenic promoters for H3K9me2 demethylation to alter chromatin accessibility in response to insulin/feeding. Our ATAC-seq clearly show that demethylation of H3K9me2 coincides with increased chromatin accessibility. Moreover, depletion of JMJD1C results in higher H3K9me2 and H3K9me3 at lipogenic promoters in the livers of JMJD1C-LKO, preventing the chromatin remodeling necessary to activate transcription. RNA-seq also confirmed downregulation of lipogenic genes in the livers of JMJD1C-LKO mice. However, gluconeogenic genes known to be suppressed in fed and activated in fasted state[37] were not affected by JMJD1C ablation, suggesting involvement of a distinct H3K9 demethylase, but not JMJD1C, for regulation of gluconeogenic genes.

Lower hepatic α-ketoglutarate level after an overnight fast and higher α-ketoglutarate level in fed mice are consistent with a previous report of decreased hepatic αKG at 12–24 h fast, although this decrease appeared to recover after a longer fasting[38]. Untargeted metabolomic studies in rats also showed lower αKG levels during fasting[39]. Our observation of insulin/feeding-induced increase in αKG level is congruent with the concept that αKG increased upon feeding/insulin treatment serves as a co-substrate for JMJD1C for demethylation of H3K9 for lipogenic gene activation. While JMJD1C demethylates H3K9me2 for transcriptional activation, there are other members of the JmjC family, such as KDM5/JARID, that demethylate H3K4 for transcriptional repression[40]. Therefore, while αKG is required for JmjC demethylases, it is not the determining factor for specific JmjC members. Demethylase activity of JMJD1C in vitro clearly show that even in the presence of an excess of αKG, JMJD1C purified from insulin-treated or refed liver, had greater activity than JMJD1C purified from serum starved/fasted liver. Regardless, αKG level was shown to be higher in hepatosteatosis and, in fact, was identified as a metabolic marker for hepatosteatosis[41].

In our present study, we provide a functional and mechanistic link between feeding/insulin-triggered mTOR signaling and downstream epigenetic modification involving JMJD1C. We show that changes in H3K9 methylation status due to mTOR-mediated phosphorylation of JMJD1C at T505 is a critical epigenetic event, leading to chromatin changes to allow lipogenesis. mTOR signaling has previously been implicated in lipogenic gene activation by inducing SREBP-1c[42]. mTOR activity is thought to increase cleavage of SREBP-1c for nuclear accumulation for lipogenic gene transcription, however, the underlying mechanism is not clear[43,44]. Our finding of phosphorylation of JMJD1C at T505 by mTOR reveals an important target and demonstrates the molecular mechanism for mTOR in lipogenic gene activation. We show that mTOR phosphorylates JMJD1C at T505, which is required for its direct interaction with USF-1 to be recruited to lipogenic promoters to allow chromatin accessibility. In response to insulin, USF-1 is phosphorylated by DNA-PK, whereas BAF60c is phosphorylated by atypical PKC to be recruited to the lipogenic promoters[10,32]. Thus, the three branches of insulin signaling pathway need to converge and the triad of DNA-PK-aPKC-mTOR underlies signaling for transcriptional activation of lipogenesis in response to insulin/feeding.

Nutrition is the principal contributory factor affecting hepatosteatosis and carbohydrate in the diet is the major source of hepatic FA production in steatotic subjects[45]. Thus, lipogenesis is a major contributing component in development of hepatosteatosis. Our present study demonstrates that depletion of JMJD1C lowered rates of lipogenesis and prevented accumulation of TG in liver thus protecting mice from development of hepatosteatosis. Interestingly, this lowered lipogenesis was met with greater accumulation of hepatic glycogen, as carbohydrate metabolites were funneled into glycogen synthesis instead. JMJD1C may provide therapeutic targets for preventing and/or controlling the development of hepatosteatosis.

The interrelationship between IR and the abnormalities of hepatic lipid homeostasis is not clear[46], since IR also accompanies abnormalities in other insulin-sensitive tissues, such as adipose and muscle. In this regard, ChREBP knockdown in the liver was shown to improve hepatosteatosis, as well as insulin sensitivity, in obese ob/ob mice[47]. In the present study, we show that liver-specific ablation of JMJD1C protects mice from hepatosteatosis and IR arising from high-CHO feeding, in the absence of increased adiposity. In population studies, a strong correlation has been reported between hepatosteatosis and IR independent of body mass index[48,49]. Our study suggests that preventing hepatosteatosis can improve IR in the absence of obesity. This implies that hepatosteatosis may lead to IR. Several GWAS studies identified JMJD1C as a prime candidate associated with plasma TG levels and VLDL size, as well as type 2 diabetes[17–21]. Interestingly, the loci identified (Chr10:63214669, Chr10:63214701, Chr10:63214700) corresponded to SNP close to T505. In conclusion, as we identify the USF-1−JMJD1C−H3K9me2 signaling axis, targeting JMJD1C phosphorylation by mTOR could provide a therapeutic strategy against hepatosteatosis and IR through perturbation of crucial lipogenic insulin signaling cascade.

## Methods

**Animals**. Animal experiments were conducted in compliance with all relevant ethical regulations for animal testing and research, and were approved by the University of California Berkeley Animal Care and Use Committee. Generation of Liver-specific JMJD1C knockout mice (JMJD1C-LKO): engineered JMJD1C mice were obtained from EMMA. These mice were bred with FLP deleter ROSA-FLPe mice (Jackson) to remove the Neo cassette, and then bred with albumin promoter driven Cre mice (Alb-Cre) (Jackson) to generate mice of liver-specific JMJD1C deletion. JMJD1C-LKO mice and their C57bl/6j WT littermates were used at 12 weeks of age unless specified otherwise. For fasting/feeding experiments, mice were fasted overnight and then fed a high CHO, fat-free diet for 8–16 h. High-CHO diet (70 kcal% carbohydrate), high-fat diet (45 kcal% fat), and high-cholesterol diet (1.5% added cholesterol) were purchased from Research Diets.

**Antibodies, cell culture, plasmid transfection**. Rabbit polyclonal-specific phospho-T505 of JMJD1C antibodies were raised against the peptide corresponding to aa 496–512 of JMJD1C (KEKFVSRPP**T**PKCVIDI) phosphorylated at T505 (P-T505) (Genemed). The specific antibody was affinity-purified using the phospho-peptide before use (diluted 1:500). The following commercial antibodies were used: JMJD1C (sc-101073, 1:500), GAPDH (sc-32233, 1:2000), USF-1 (sc-229, 1:1000), HDAC1 (sc-7872, 1:1000), Fatty acid synthase (sc-55580, 1:1000), and SREBP-1c (sc-366, 1:1000) from SCBT. FLAG (147935, 1:1000), HA (C29F4, 1:1000), and Raptor (24C12, 1:750) from CST. phospho-Threonine (05-1923, 1:200), phosphor-Serine (Ab1603, 1:200), and normal IgG (PP64B, 2 μg/μg of chromatin) from EMD Millipore. GFP (ab1218, 1:1000), Total H3 (ab1791, 2 μg/μg of chromatin), H3K9me1 (ab9045, 2 μg/μg of chromatin), H3K9me2 (ab176882, 2 μg/μg of chromatin), H3K9me3 (ab8898, 2 μg/μg of chromatin), and H3K4me3 (ab8580, 2 μg/μg of chromatin) from Abcam. Full-length human JMJD1C plasmid was obtained from OriGene.

HepG2 and 293FT cells were obtained from ATCC. Cells were grown in Dulbecco's modified Eagle medium (DMEM) supplemented with 10% fetal bovine serum and 100 units/mL of penicillin and streptomycin. HepG2 cells were maintained in serum-free media overnight prior to treatment with 100 nM insulin for 30 min. HEK293FT cells in DMEM were supplemented with 10% fetal bovine serum and 100 units/mL penicillin, streptomycin, and neomycin, and 293F cells in 293 Freestyle media were transfected with plasmids using Lipofectamine 2000

(Invitrogen). Adenovirus containing JMJD1C, sh-JMJD1C, and JMJD1C mutants (T505A and T505D) were generated by Vector Biolabs and were used for infecting HepG2 cells or for tail-vein injection into mice.

**Mass spectrometry analysis**. To identify USF-1 interacting proteins we employed tandem affinity purification (TAP), using TAP-tagged USF-1 as bait, and incubated USF-1 with liver nuclear extracts prior to submission for mass spectrometry analysis. Site-specific phosphorylation of JMJD1C was detected by enzymatically digesting HA-purified JMJD1C from HepG2 nuclear extracts and subjecting digested peptides to two-dimensional "MudPIT" run (cation exchange/reversed-phase liquid chromatography–tandem MS) using a Thermo LTQ XL mass spectrometer. DTASelect program was used to interpret the mass spectra.

**JMJD1C activity and α-ketoglutarate levels**. JMJD1C was overexpressed in HepG2 cells by JMJD1C-HA adenovirus. After treatment with 100 nM insulin for 30 min, JMJD1C was purified from cell lysates of HepG2 cells and purified using HA beads. In vitro demethylation assay was performed using ELISA kits containing methylated H3K9 (Active Motif and Epigentek), according to the manufacturer's instructions. To determine substrate specificity, we incubated purified JMJD1C with individual substrate, H3K9me3, H3K9me2, or H3K9me1 for 60 min. After primary and secondary antibody treatments, samples were incubated with the developing solution before measuring absorbance at 450 nm on a spectrophotometer. Remaining substrate levels were calculated using the standard curve.

Total JMJD1C demethylase activity was measured using H3K9 Specific demethylase kit (Epigentek, P-3077-48). Briefly, JMJD1C was purified from JMJD1C-HA overexpressing HepG2 cells with or without 100 nM insulin treatment. JMJD1C was incubated with substrate and co-factor in assay buffer for 60 min. After washing, fluorescence developing solution was added and fluorescence intensity was measured by excitation at 530 nm and emission at 590 nm. For hepatic JMJD1C activity, nuclear proteins were extracted from livers of mice that were fasted or fed. JMJD1C was purified by using JMJD1C antibody and protein A/G Plus agarose (Thermo-Fisher), before subjecting them to demethylation assay.

α-ketoglutarate levels were measured by using an α-ketoglutarate assay kit (Abcam, ab83431). Briefly lysates of cells or tissues were incubated with the enzyme reaction mixture for 30 min at 37 °C for transamination of α-ketoglutarate with the generation of pyruvate that was utilized to convert a probe to a color, which was read by spectrophotometer at 570 nm. α-ketoglutarate concentration was calculated by using a standard curve.

**ChIP–qPCR and ChIP-seq**. ChIP was performed using ChIP kit (Cell signaling, 57976s). Briefly, HepG2 cells with or without insulin treatment and livers from fasted or fed mice were fixed with disuccinimidyl glutarate (DSG) at 2 mM concentration in PBS for 45 min at room temperature before cross-linking with 1% formaldehyde in PBS for 10 min. The reaction was stopped by incubating with 125 mM glycine for 10 min. Cells or tissues were rinsed with ice-cold phosphate-buffered saline (PBS) for three times, and lysed in IP lysis buffer containing 500 mM HEPES-KOH, pH 8, 1 mM EDTA, 0.5 mM EGTA, 140 mM NaCl, 0.5% NP-40, 0.25% Triton X-100, 10% glycerol, and protease inhibitors for 10 min at 4 °C. Nuclei were collected by centrifugation at $600 \times g$ for 5 min at 4 °C. Nuclei were released by douncing on ice and collected by centrifugation. Nuclei were then lysed in nuclei lysis buffer containing 50 mM Tris, pH 8.0, 1% SDS 10 mM EDTA supplemented with protease inhibitors, and sonicated three times by 20 s burst, each followed by 1 min cooling on ice. Chromatin samples were diluted 1:10 with the dilution buffer containing 16.7 mM Tris pH 8.1, 0.01% SDS 1.1 % Triton X-100 1.2 mM EDTA, 1.67 mM NaCl, and proteinase inhibitor cocktail. Soluble chromatin was quantified by absorbance at 260 nm, and equivalent amounts of input DNA were immunoprecipitated using 10 μg of indicated antibodies or normal mouse IgG (Santa Cruz) and protein A/G magnetic beads (Thermo-Fisher). After the beads were washed and cross-linking reversed, DNA fragments were purified using QIAquick DNA purification kit (Qiagen). The promoter occupancy for FAS and other indicated genes was confirmed by qPCR. The fold enrichment values were normalized to input.

For ChIP-seq, we used two mice per condition and processed chromatin separately, thus each sample analyzed was a separate biological replicate. Library preparation and ChIP-seq were performed with eluted DNA using an Illumina Hiseq4000 sequencer by the QB3 Berkeley Genomics Lab (Berkeley, CA). Partek Flow Genomics Suite was used to analyze sequencing data. Reads were aligned to mm10 genome assembly using Bowtie2 and only reads aligned to a unique genome location were reported. The aligned reads were then subjected to peak calling using MACS2 with a q-value cutoff of 0.05. Reads with 2.5-fold enrichment compared to the input sample, were considered as peaks. Peaks were annotated by overlapping to RefSeq mRNA database and visualized using UCSC genome browser. Gene set enrichment and pathway analysis was used to highlight biological processes with the highest q-values among those identified.

**ATAC-seq**. For ATAC-seq, we used two mice per condition and processed nuclei extracted from liver separately, thus each sample analyzed was a separate biological replicate. ATAC-seq was performed using Nextera DNA library Preparation kit

(Illumina, 15028212) according to Chen et al. [50]. In all, $5 \times 10^4$ nuclei from cells or tissues were collected in lysis buffer containing 10 mM Tris pH 7.4, 10 mM NaCl, 3 mM MgCl$_2$, and 1% NP-40, and spun at $500 \times g$ at 4 °C for 10 min. The pellets were resuspended in the transposase reaction mixture containing 25 μL 2 × Tagmentation buffer, 2.5 μL transposase, and 22.5 μL nuclease-free water, and incubated at 37 °C for 30 min. The samples were purified using MinElute PCR Purification kit (Qiagen, 28006) and amplification was performed in 1 × next PCR master mix (NEB, M0541S) and 1.25 μM of custom Nextera PCR primers 1 and 2 with the following PCR conditions: 72 °C for 5 min; 98 °C for 30 s; and thermocycling at 98 °C for 10 s, 63 °C for 30 s, and 72 °C for 1 min. Samples were amplified for five cycles and 5 μL of the PCR reaction was used to determine the required cycles of amplification by qPCR. The remaining 45 μL reaction was amplified with the determined cycles and purified with MinElute PCR Purification kit (Qiagen, 28006) yielding a final library concentration of ~30 nM in 20 μL. Libraries were subjected to pair-end 50 bp sequencing on HiSeq4000 with 4–6 indexed libraries per lane. Partek Flow Genomics Suite was used to analyze sequencing data. Reads were aligned to mm10 genome assembly using BWA-backtrack. MACS2 in ATAC mode was then used to identify peaks from the aligned reads with a q-value cutoff of 0.05 and fold enrichment cutoff of 2.0. Quantify regions tool was used to quantify the peaks identified by MACS2 from each sample to generate a union set of regions. Regions were then annotated to RefSeq mRNA database and analysis of variance (ANOVA) was performed to determine significant peak differences between groups. UCSC genome browser was used to visualize peaks. Gene set enrichment and pathway analysis was used to highlight biological processes with the highest q-values among those identified.

**Glycogen staining and measurement**. Periodic acid–Schiff (PAS) staining kit for glycogen was purchased from Sigma. Briefly, paraffin-embedded liver and muscle sections were deparaffinized and hydrated with water. Oxidized in 0.5% periodic acid solution for 5 min and rinsed in distilled water. Sections were placed in Schiff reagent for 15 min, washed in lukewarm tap water for 5 min followed by Mayer's hematoxylin counterstaining for 1 min, washed in tap water for 5 min and dehydrated, and coverslipped using a synthetic mounting medium. Glycogen levels in liver and muscles of the mice were using glycogen assay kit (sigma). Briefly, 10 mg of tissues were homogenized in 100 μl water and boiled for 5 min. Homogenates were clarified by centrifugation at $13,000 \times g$ for 5 min. Supernatant was used for the assay. Samples were diluted to final volume of 50 μL with hydrolysis buffer. Samples together with standards were added with 2 μL of hydrolysis enzyme mix, and incubated at room temperature for 30 min. Then developed by adding 50 μL master reaction mix containing development enzyme mix, buffer, and peroxidase substrate. For colorimetric assay, the absorbance was measured at 570 nm. Concentration of glycogen were calculated by using the standard curve.

**In vitro phosphorylation**. mTOR complex components, Myc-mTOR (Addgene, 1861), HA-Raptor (Addgene, 8513), and Flag-Rheb (Addgene, 19996), were transfected individually into HEK293 cells grown in 100 mm dishes ($2 \times 10^6$ cells), and each component was purified from cell lysates using Myc, HA, and Flag antibodies. GST-JMJD1C purified after expressing it in E. coli was incubated with mTOR complex (1:1:1 ratio of total 300 ng in 15 μL) in the kinase buffer containing 50 mM HEPES, pH 7.4, 10 mM MgCl$_2$, 10 mM MnCl$_2$, 1 mM DTT with or without 500 μM ATP at 25 °C for 1 h and the reaction was terminated by adding 20 μL 2 × SDS sample buffer. Phosphorylation reaction mixture containing 0.5 μg of protein was separated by sodium dodecyl sulfate–polyacrylamide gel electrophoresis (SDS-PAGE) and JMJD1C phosphorylation was detected with pan-phospho-serine and pan-phospho-threonine antibodies or phospho-specific anti-peptide antibody that we have raised as indicated above.

**Immunoprecipitation, GST pull-down, and luciferase reporter assays**. For immunoprecipitation, nuclear extracts were incubated with the specific antibodies overnight at 4 °C followed by protein G agarose beads (Santa Cruz) and the immunoprecipitates were separated by SDS-PAGE. Proteins were transferred onto nitrocellulose membranes (BioRad) for immunoblotting. For GST pull-down, bacterially expressed GST-USF-1 fusion proteins on glutathione-agarose beads (Santa Cruz) were incubated with $^{35}$S labeled in vitro transcribed and translated proteins. The complex formed was separated by SDS-PAGE before autoradiography. The 293FT cells were transfected with-444-FAS-Luc along with various expression constructs using Lipofectamine 2000 (Invitrogen) and luciferase assays were performed using Dual-Luc reagent (Promega).

**Quantitative reverse transcription PCR (RT-qPCR)**. One microgram of total RNA isolated using Trizol reagent (Gibco-BRL) were reverse transcribed and the resultant cDNAs were amplified by qPCR using 7500 Fast Real-time PCR system (ABI). The relative mRNA levels were quantified using GAPDH as control. Primer sets for FAS, mGPAT, ACC, SCD1, and SREBP-1c, were from Elimbio. Statistical analysis of the qPCR was obtained using the TMCt method.

**Preparation of Nascent RNA**. Livers from three mice were homogenized in five volumes of buffer containing 0.32 M sucrose, 3 mM MgCl$_2$, 5 mM Hepes (pH 6.9), and 0.5 mM 8-mercaptoethanol. Nuclei collected by centrifugation were washed

once by centrifugation through a 2.1 M sucrose cushion at 20,000 rpm for 60 min in a Beckman SW 28 rotor. The nuclei were stored in liquid nitrogen in 50 mM Tris (pH 7.9), 5 mM MgCl$_2$, 0.5 mM 8-mercaptoethanol, and 40% glycerol. Nuclei were treated with DNase (Roche) and RNA were purified using RNeasy kit (Qiagen).

**Measurement of rate of de novo lipogenesis**. Fatty acids synthesized during a 24-h $^2$H$_2$O body water-labeling period were measured[51]. Mass Isotopomer Distribution Analysis (MIDA) was performed. Fractional DNL contribution was calculated as previously described by $f$DNL = EM1FA/A[52].

**Statistical analysis**. Statistical comparisons were made using a two-tailed unpaired $t$-test using GraphPad Prism 8 software (GraphPad Software Inc., La Jolla, CA, USA). For genome-wide analyses, we employed Partek Genomics Suite (Partek Inc., St. Louis, Missouri, USA) using ANOVA for ATAC- and ChIP-seq comparisons, and DESeq2 for RNA-seq differential expression comparisons, and subsequently used Gene set enrichment tool for gene ontology to identify the significantly affected pathways presented in Fig. 5.

**Reporting summary**. Further information on research design is available in the Nature Research Reporting Summary linked to this article.

## Data availability

The genomic and transcriptomic datasets generated and analyzed during the current study are available in the GEO repository under accession number: GSE142815. The proteomic datasets generated and analyzed during the current study are available in the MassIVE repository under accession number MSV000084745.

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

## Acknowledgements

This work was supported by DK081098 to H.S.S. and DK105671 to J.A.V. The mass spectrometric analysis was performed at the Vincent J. Proteomics/Mass Spectrometry Laboratory at UC Berkeley, supported by S10RR025622.

## Author contributions

J.A.V., Y.W. and H.S.S. designed the study; J.A.V., Y.W. and H.N. conducted experiments; Y.C. assisted in performance of sequencing and data analysis; J.A.V., Y.W. and H.S.S. interpreted data and wrote the paper; all authors discussed the results and approved the final manuscript.

## Competing interests

The authors declare no competing interests.
