## [Peer Review File · Nature Communications]

Reviewers' comments:

Reviewer #1 (Remarks to the Author):

Viscarra et al describe studies on the role of JMJD1C in mouse liver during the feeding cycle. They first present MS analysis of proteins that interact with USF1 and peptides corresponding to JMJD1C show up in a tandem affinity pulldown of USF1. Then studies identify the bHLH domain of USF1 and the amino-terminal domain of JMJD1A interact through in vitro pull downs and transient assays demonstrate the Fasn promoter is activated in a concerted manner by transfection of JMJD1C and USF1 in HepG2 cells. Then they show by ChIP that JMJD1C is recruited to promoters of lipogenic genes. Studies in animals, first with adenovirus over-expression and sh knockdown, then using a liver knockout model for JMJD1C show JMJD1C in liver is required for lipogenic gene activation, TG accumulation and increased DNL during feeding and in response to direct administration of insulin. JMJD1CLKO mice gain weight on High Carb or high fat diet similar to WT mice but are protected from glucose intolerance suggesting a disconnect between weight gain and glucose regulation, a phenomenon reported in several other settings as well.

Mechanistically, the studies provide evidence that JMJD1C directly de-methylates Lys9me2 in proximal promoters during feeding/insulin treatment through a direct TORC1 mediated phosphorylation of JMJD1C. The basic observation that JMJD1C is a newly identified regulator of hepatic lipogenesis is clear from the studies but the mechanism proposed is simplistic, does not clearly consider data on CHREBP, SREBP-1, and LXR in the literature for a comprehensive model and studies are not well controlled. A reliance on a small region of the proximal promoter of one gene in the lipogenic cascade is a very narrow focus and likely not a general mechanism for JMJD1C's more general role. There is critical information missing from the methods section, many controls are missing and the descriptions of the genome wide approaches are not detailed enough for careful evaluation and the presentation lacks overall experimental rigor.

1). Whether JMJD1C recruitment to more than one lipogenic gene is not clear and whether USF1 is required in vivo at all has not been established. whether USF factors are general activators of other lipogenic genes has not been established. Thus, the recruitment of JMJD1C to lipogenic gene promoters through USF1 is probably not a general mechanism. How is JMJD1C recruited to other lipogenic gene promoters?

2). Whether JMJD1C is phosphorylated by TORC1 specifically in vivo has not been clear-rapamycin or raptor KO needs to be pursued. The combination of TORC subunits used does not represent the complete TORC1 complex so specificity in vitro is also questionable.

3). There appears to be no description of the Mass Spectrometry in the methods section-either for the identification of USF1 interacting proteins or for the phospho amino acid analysis of JMJD1C.

4). Eventhough the JMJD1C LKO results in reduced TG in the liver, the level of ~150 mg/g in the JMJD1CLKO is still very high and likely contributes to impaired glucose regulation. This would need a chow fed control for both HFD and high carb. to be rigorous. Also, it is not clear why the glucose levels for the GTT and ITT in Fig. 3B have different starting values? Also, an AUC analysis would likely show no ITT difference?

5). The histone methylation mark ChIP studies need a total H3 ChIP for control to ensure equal H3 levels-without this it is impossible to be sure difference in methylation marks are due to altered methylation specifically or altered H3.

6). In this regard, authors should be aware that h3K9me1 is a histone mark associated with activation

7). There is not enough description of ChIP-seq and ATAC-seq to carefully assess the data. There is no description of how peaks are identified by ATAC-seq. The relationship of the peaks discussed in Fig. 5 to the overall collection of peaks genome wide is not clear. Text is written to suggest most

ATAC-seq peaks are promoter proximal whereas this is unlikely based on other ATAC-seq genome wide studies. Same is true for H3K9me2. Also, comparison of H3K9me2 genome wide in Fast/refeed would be helpful to compare for author's major hypothesis. How were genes chosen for TSS analysis in Fig. 5A and B? How ATAC-seq and CHIP-seq samples were analyzed needs to be included; were replicates performed-how consistent?, were samples pooled from more than one mouse, how many mice analyzed?, how were the peak differences quantified? Box plots and/or violin plots showing read density comparisons and statistics need to be included to assess the peak differences in the comparative data sets. Were scales in Genome browser shots the same? These are need to compare peak heights. RNA seq comparison in Fig. 5C needs more full description of the exact comparison being made.

8). The dramatic difference in alpha-ketoglutarate levels is surprising and could have more profound implications for other JMJD family proteins as well as for TCA cycle flux and glutamate metabolism. How does the measurement of alpha-ketoglutarate levels compare with other reports in the literature for fast/feeding or even circadian variation where lipogenesis also varies?

9). JMJD1C protein expression loading control needed in Fig. 7C to ensure T505A is not expressed at lower level

10). References to "previously described" methods need to have references associated

Reviewer #2 (Remarks to the Author):

In this manuscript, Viscarra et al. conducted a series of experiments to show that JMJD1C is recruited by USF-1 to various lipogenic genes for H3K9 demethylation to enhance chromatin accessibility in the fed state. This is achieved via mTORC1-mediated phosphorylation of JMJD1C on threonine 505 and increased availability of the co-substrate α -ketoglutarate in response to insulin or feeding.

Based on their observations, the authors infer that their "*results provide evidence that hepatosteatosis, rather than adipose expansion or the size of adipose tissue depots, may be more critical in the development of insulin resistance that accompanies diet-induced obesity*". The authors provide novel insights on regulatory mechanisms of hepatic lipogenesis that are relevant to hepatosteatosis. There are a few concerns that the authors should consider carefully.

Major concerns

1. H3K9me1 has been shown as histone mark of distinct regions of silent chromatin (*J Biol Chem* 2006 May 5;281(18):12760-6; *Biochimie* Vol 94 (12), December 2012, pp 2656-2664; *Experimental & Molecular Medicine* vol 49, p e324 (2017); *Genes Dev.* 2002 Jul 15;16(14):1779-91.). The authors cite a paper suggesting that H4K9me1 is found in the promoter regions of active genes (*Cell* Vol 138, (5), 2009, pp 1019-1031). This controversy should be properly discussed. In any case, since HepG2 cells treated with insulin and mice refeed after a fasting period show increased H3K9me1 levels on lipogenic genes whose expression increases under such conditions, how do the authors explain this observation? How can it be that the increase of a mark of silent chromatin leads to increased expression of lipogenic genes? Is it possible that, depending on the chromatin/promoter context, H3K9me1 may act either as mark of silent or active genes? Also, H3K4me3, a mark of active promoters increases under the same conditions. Is it possible that H3K4me3 is a stronger epigenetic mark than H3K9me1? The authors should provide evidences of how these two histone marks play a role in the expression of lipogenic genes. Moreover, how do the authors explain that H3K9me1 levels increase under insulin or fed conditions when JMJD1C should demethylate H3K9me1?

2. The quality of western blot in Figure 6A right is poor: the band of fasted mice total JMJD1C is covered with a white spot, likely due to uneven blotting and transfer in this part of the gel.

3. Likewise, the WB images of co-IP experiment in Figure 7G lower panel is poor: no band are

visible in input samples; also, the bands in the IP of USF-1 are quite faint! This experiment should be repeated to provide a more convincing image of the bands.

4. The authors claim that their "*results provide evidence that hepatosteatosis, rather than adipose expansion or the size of adipose tissue depots, may be more critical in the development of insulin resistance that accompanies diet-induced obesity*". I believe that this statement is a bit strong and I suggest to tone it down. A study with longer treatment with HFD should be performed to corroborate the view that hepatosteatosis, rather than adipose tissue expansion or size of adipose tissue depots, may be more critical in the development of insulin resistance following diet-induced obesity; with the present data it cannot be ruled out that adipose tissue expansion contributes to insulin resistance. It is likely that both hepatosteatosis and adipose tissue expansion contribute together, in a concerted and stepwise fashion, to the onset of insulin resistance. For instance, it cannot be ruled out that some signal originating from adipose tissue in mice treated with HFD (e.g., proinflammatory cytokines that are released by WAT) may trigger hepatosteatosis. At this stage of knowledge, I would be cautious to overstate the primary importance of hepatosteatosis and rule out the role of WAT in the onset of insulin resistance.

Minor concerns

1. Figure 3A middle panel shows results of JMJD1C gene expression in brain liver and kidney. The same results seem to be reported in Figure S2 panel A with the addition of JMJD1C gene expression in the lung, heart, spleen and muscle. I suggest to simply replace Figure 3A middle panel with Figure S2 panel A. Of course, Figure S2A should be removed from Figure S2.

2. Pg 11, line 226: the sentence "*demonstrating that JMJD1C protected these mice from hepatosteatosis*" should be corrected as follows: "*demonstrating that **knock out of JMJD1C in the liver** protected these mice from hepatosteatosis*". Also, the following sentence "*Glycogen levels upon refeeding were again significantly higher in the JMJD1C-LKO livers, compared to WT mice (Fig. 3B bottom left)*" should be "*Glycogen levels upon refeeding were again significantly higher in the JMJD1C-LKO livers, compared to WT mice (Fig. 3B **middle right**)*".

3. Figure S2C left panel: images of different fat depots are not labelled. Presumably, as stated in the figure legend, subWAT are the images right below the liver images, epi-WAT the next ones and BAT the images at the bottom of the panel, however I recommend labelling the images directly in the panel. Further, in the right panel the quantification of WAT refers to subWAT, epi-WAT or to the sum of both? Please, specify.

Reviewer #3 (Remarks to the Author):

In this original work submission Viscarra and colleagues characterize the physiologic role and mechanisms of the histone demethylase JMJD1C in hepatic insulin responses. The authors postulate that JMJD1C undergoes site-specific phosphorylation by mTOR and is recruited to lipogenic gene promoters in response to fast/refeed. They further surmise that JMJD1C demethylates specific histone marks to alter chromatin dynamics at lipogenic genes.

The work utilizes unbiased approaches that build logically on multiple previous publications from this group and adds to our understanding of mechanisms orchestrating promoter activation in metabolic control. The text is well-written although figure renderings could be improved. There are a number of issues with the proposed mechanism and other technical considerations that should be addressed but on balance this is a novel and timely story.

1. The authors state that collaborative interaction of transcription factors at lipogenic promoters in response to insulin require an accessible chromatin landscape.

One would presume then that overexpression of nuclear SREBP1c may not rescue the phenotype of JMJD1C liver specific knockout since chromatin architecture may be limiting (would be a highly striking finding). Is this the case? The authors should administer control virus and AAV (or adeno) SREBP1c in WT and L-KO to test this directly. Although convincing evidence is presented that

JMJD1C contributes to epigenetic landscape and transcriptional outputs at lipogenic genes, the epistatic relationship between JMJD1C and canonical transcriptional regulators in insulin responses is not established and these experiments will provide critical insight.

2. Lipogenic genes are regulated in response to dietary clues through diverse mechanisms but the notion that JMJD1C induces lipogenic genes ONLY in response to fast-refeed is not convincingly established. The nuclear receptors LXRs are direct inducers of lipogenesis in response to cholesterol overload in addition to contributing to fast/refeed responses. Does the administration of LXR agonists (readily available commercially) in the setting of JMJD1C loss of function show differential regulation of lipogenic genes? How about administration of cholesterol rich diet? The studies will clarify the specificity of JMJD1C effects on lipogenic genes.

3. Insulin and fast refeed responses activate lipogenic genes in addition to inhibiting gluconeogenesis (PMID 20133650). Does loss of JMJD1C influence the expression of gluconeogenic genes in response to insulin and fast/refeed? The authors should show clear qPCR analysis since the ATAC-seq of gluconeogenic genes in S3 are difficult to interpret (based on in vivo data it appears that JMJD1C strongly influences glucose responses and figure 4C suggests that JMJD1C may perhaps have additional effects on chromatin besides demethylating H3K9me2 and 1). If JMJD1C does influence gluconeogenesis how do the authors consolidate previous findings showing that insulin suppression of gluconeogenesis does NOT require mTOR? Should be addressed in discussion.

4. Did the authors perform genome-wide normalization for ATAC-seq data for L- JMJD1C KO? The background appears to be much higher in L- JMJD1C KO samples so would be cautious in making the claim that access sites show differential openness. Did the authors perform motif analysis on differential comparing WT and L- JMJD1C KO refeed?

Minor

1. Please provide more details on ATAC seq analysis. For example what fold cutoff was used for accessible peaks or did you use a specific statistical test?

2. The authors should show protein blots for a number of lipogenic genes related to figures 2 and 3. It appears that changes in gene expression are substantially more dramatic than observed phenotype.

3. The genetic background for the JMJD1C flox mice is presumably bl6. Please confirm in methods since this is not clear from the checklist.

4. The figures and figure legends are frustratingly difficult to follow with excessive use of right, left, middle top etc... For many figure groupings break the panels into individual figures using additional letters. For example see panels figure 2A , 2C, and 2D.

5. The paper relies heavily on 293 and HepG2 cells which do entirely recapitulate in vivo responses and for this reason not used extensively in previous investigations of hepatic insulin responses. Suggest discussing in limitations.

Reviewer #1

Viscarra et al describe studies on the role of JMJD1C in mouse liver during the feeding cycle. They first present MS analysis of proteins that interact with USF1 and peptides corresponding to JMJD1C show up in a tandem affinity pulldown of USF1. Then studies identify the bHLH domain of USF1 and the amino-terminal domain of JMJD1A interact through in vitro pull downs and transient assays demonstrate the Fasn promoter is activated in a concerted manner by transfection of JMJD1C and USF1 in HepG2 cells. Then they show by ChIP that JMJD1C is recruited to promoters of lipogenic genes. Studies in animals, first with adenovirus over-expression and sh knockdown, then using a liver knockout model for JMJD1C show JMJD1C in liver is required for lipogenic gene activation, TG accumulation and increased DNL during feeding and in response to direct administration of insulin. JMJD1CLKO mice gain weight on High Carb or high fat diet similar to WT mice but are protected from glucose intolerance suggesting a disconnect between weight gain and glucose regulation, a phenomenon reported in several other settings as well. Mechanistically, the studies provide evidence that JMJD1C directly de-methylates Lys9me2 in proximal promoters during feeding/insulin treatment through a direct TORC1 mediated phosphorylation of JMJD1C.

The basic observation that JMJD1C is a newly identified regulator of hepatic lipogenesis is clear from the studies but the mechanism proposed is simplistic, does not clearly consider data on CHREBP, SREBP-1, and LXR in the literature for a comprehensive model and studies are not well controlled. A reliance on a small region of the proximal promoter of one gene in the lipogenic cascade is a very narrow focus and likely not a general mechanism for JMJD1C's more general role. There is critical information missing from the methods section, many controls are missing and the descriptions of the genome wide approaches are not detailed enough for careful evaluation and the presentation lacks overall experimental rigor.

In our present study, we specifically addressed JMJD1C recruitment by USF-1 and its phosphorylation by mTORC to activate lipogenic genes in response to insulin/refeeding. By ChIP-qPCR we demonstrated JMJD1C recruitment to promoter regions of various lipogenic genes (Fig. 1). Moreover, by ChIP-seq of H3K9me2 and RNA-seq, we showed that JMJD1C recruitment changes chromatin at the genome-wide level, which include various lipogenic genes. In this revision, we include description of our mass spec analyses, additional controls for our ChIP-qPCR, as well as controls, such as chow diet feeding in animal studies. We also include more detailed descriptions and analysis for better evaluation of the genome-wide data.

Moreover, as suggested by the reviewer, we performed additional experiments to address the potential recruitment of JMJD1C by ChREBP, SREBP-1 or LXR by co-IP and in vitro interaction assays. Co-IP of lysates from 293FT cells overexpressing ChREBP, SREBP-1c, or LXR detected interaction of JMJD1C with SREBP-1c and LXR, but not with ChREBP (Fig. S1A). However, our pull-down of recombinant JMJD1C with purified SREBP-1c or LXR did not detect interaction of JMJD1C with SREBP-1c or LXR (Fig. S1B). These results indicate that, although SREBP1 or LXR can be found in JMJD1C complex, JMJD1C does not directly interact with SREBP-1c or LXR. We previously reported that USF-1 directly interacts with SREBP1 to recruit it to the promoter regions of lipogenic genes (Griffin et al. J Bio Chem 2007) and thus via direct interaction with USF-1, SREBP1 could be found in complex containing JMJD1C. As for LXR, further studies would be needed to understand how LXR can indirectly interact with JMJD1C. In this revision, we include and discuss these additional results, to provide a more general picture how JMJD1C functions in lipogenic gene activation.

1). Whether JMJD1C recruitment to more than one lipogenic gene is not clear and whether USF1 is required in vivo at all has not been established. whether USF factors are general activators of other lipogenic genes has not been established. Thus, the recruitment of JMJD1C to lipogenic gene promoters through USF1 is probably not a general mechanism. How is JMJD1C recruited to other lipogenic gene promoters?

As mentioned above, we previously reported that USF-1 binds E-box on various lipogenic promoter regions and recruits SREBP-1c and other cofactors to activate various lipogenic genes (Griffin et al. J Biol Chem 2007, Latasa et al. PNAS 2000, Wong et al. Cell 2009, Wong et al. Cur Opin Pharma 2010, Wang et al. Mol Cell 2013, Wang et al. Nat Rev Mol Cell Bio 2015, Viscarra et al. Sci Sig 2017). Moreover, we tested multiple lipogenic gene promoter regions by ChIP-qPCR (Fig. 1) showing that JMJD1C is recruited to the promoter regions of various lipogenic genes. We also examined by ChIP-seq for H3K9Me2, as well as RNA-seq of JMJD1C-LKO livers that showed effects on numerous lipogenic genes. We also performed

motif analysis comparing differential motifs between our WT and JMJD1C-LKO ATAC-seq datasets and found that while the motif for USF-1 was significantly enriched in the WT but not in JMJD1C-LKO, providing further evidence of JMJD1C recruitment by USF-1.

2). Whether JMJD1C is phosphorylated by TORC1 specifically in vivo has not been clear-rapamycin or raptor KO needs to be pursued. The combination of TORC subunits used does not represent the complete TORC1 complex so specificity in vitro is also questionable.

As suggested by the reviewer, we treated mice with rapamycin and found that rapamycin treatment completely blocked JMJD1C T505 phosphorylation that we detected upon refeeding (Fig. 7F). We also performed knockdown of Raptor by adenoviral injection into tail vein of mice. We detected greatly impaired phosphorylation of JMJD1C at T505 in refed mice, when Raptor level was reduced (Fig. 7G). We also detected impairment of lipogenic gene induction by rapamycin treatment or KD of Raptor, that blocked T505 phosphorylation (Fig. 7F-G). These results clearly demonstrate that mTORC1 phosphorylates JMJD1C at T505, and that this phosphorylation event is necessary for lipogenic gene induction in response to feeding/insulin.

3). There appears to be no description of the Mass Spectrometry in the methods section-either for the identification of USF1 interacting proteins or for the phospho amino acid analysis of JMJD1C.

We thank the reviewer for pointing this out, we have updated the methods section to include a description of the mass spectrometry methods we used for interaction and phosphorylation analyses.

4). Eventhough the JMJD1C LKO results in reduced TG in the liver, the level of ~150 mg/g in the JMJD1CLKO is still very high and likely contributes to impaired glucose regulation. This would need a chow fed control for both HFD and high carb. to be rigorous. Also, it is not clear why the glucose levels for the GTT and ITT in Fig. 3B have different starting values? Also, an AUC analysis would likely show no ITT difference?

As suggested by the reviewer, we now include data obtained from mice on chow diet for 4 months (Fig. S2A-C) to compare with our high CHO and high fat diet studies. Chow-fed WT mice had hepatic TG levels of 252 mg/g and JMJD1C-LKO mice had 127 mg/g (Fig. S2C). However, GTT and ITT (by AUC analysis also) showed that WT (as well as JMJD1C-LKO) mice were still not insulin resistant, not making it possible for us to demonstrate improvement of insulin resistance by JMJD1C deficiency (Fig. S2A).

Upon high fat diet feeding, both WT and JMJD1C-LKO mice showed higher adiposity. However, JMJD1C-LKO mice had lower hepatic TG levels and these KO mice were protected from obesity-related insulin resistance.

Upon high CHO diet feeding, JMJD1C-LKO mice again had significantly lower liver TG levels compared WT mice. High CHO diet feeding caused insulin resistance in WT mice, whereas JMJD1C-LKO remained insulin sensitive, prevented from high CHO diet-induced insulin resistance. Importantly, neither WT nor KO mice were obese and they showed similar fat pad weights to chow feeding. Therefore, we conclude that JMJD1C prevents high CHO diet-induced hepatosteatosis and insulin resistance by JMJD1C ablation in the absence of adiposity. We now state that JMJD1C prevents high CHO diet-induced hepatosteatosis and insulin resistance, in the absence of any changes in adiposity by JMJD1C ablation (Fig. S2A-B).

Prior to performing the GTTs, we fasted mice overnight before giving a glucose bolus, as is typically done. Prior to performing ITT we only fasted mice for 4 hrs before the insulin injection, as is also a standard method. Therefore, starting values for glucose were different at time zero. We added AUC analysis for all GTT and ITT showing significant difference for both high CHO diet and high fat diet (Fig. 3 and Fig. S2H),

5). The histone methylation mark ChIP studies need a total H3 ChIP for control to ensure equal H3 levels-without this it is impossible to be sure difference in methylation marks are due to altered methylation specifically or altered H3.

As suggested by the reviewer, we now include data from ChIP experiments for total H3 showing the levels of H3 do not significantly change in either cell culture or animal studies and thus changes in methylation marks are due to altered methylation and not total H3 levels (Fig. 4).

6). In this regard, authors should be aware that h3K9me1 is a histone mark associated with activation.

As indicated by the reviewer, the specific function of H3K9me1, whether silencing or activating, remains controversial. In fact, the reviewer 2 also pointed out this controversy. "H3K9me1 has been shown as histone mark of distinct regions of silent chromatin (*J Biol Chem* 2006 May 5;281(18):12760-6; *Biochimie* Vol 94 (12), December 2012, pp 2656-2664; *Experimental & Molecular Medicine* vol 49, p e324 (2017); *Genes Dev.* 2002 Jul 15;16(14):1779-91.), while other papers suggest that H3K9me1 is found in the promoter regions of active genes (*Cell* Vol 138, (5), 2009, pp 1019-1031)." We observed accumulation of H3K9me1, correlating with lipogenic gene activation. However, it is possible that accumulation of H3K9me1 was due to the fact that JMJD1C has higher demethylase activity for H3K9me2 than H3K9me1. Thus, H3K9me2 was demethylated by JMJD1C more efficiently than H3K9me1, resulting in H3K9me1 accumulation in lipogenic gene activation. We include discussion on this controversy in the text.

7). There is not enough description of ChIP-seq and ATAC-seq to carefully assess the data. There is no description of how peaks are identified by ATAC-seq. The relationship of the peaks discussed in Fig. 5 to the overall collection of peaks genome wide is not clear. Text is written to suggest most ATAC-seq peaks are promoter proximal whereas this is unlikely based on other ATAC-seq genome wide studies.

As suggested by the reviewer, we edited the methods section to better describe our ATAC- and ChIP-seq experiments. We have added a description of how ATAC peaks were identified and why we chose to highlight those peaks shown in Fig. 5. We have edited the text to better describe where peaks were found genome-wide. We also include gene section breakdown analysis for our ATAC-seq. Approximately 38% of the peaks were found near the TSS, 25% were found in intragenic regions, and approximately 37% of peaks were found in the intergenic regions (Fig. S6), which is similar to previously reported ATAC-seq analyses (Ackermann et al. *Mol Metab* 2016, Philip et al. *Nature* 2017, Liu et al. *Sci Data* 2019).

Same is true for H3K9me2. Also, comparison of H3K9me2 genome wide in Fast/refeed would be helpful to compare for author's major hypothesis. How were genes chosen for TSS analysis in Fig. 5A and B?

We edited the text to more clearly describe where H3K9me2 peaks were found genome-wide. We also include gene section breakdown analysis for H3K9me2 ChIP-seq showing approximately 47% of peaks at the TSS, 49% in intragenic regions, and only 4% in the intergenic regions (Fig. S6), similar to previously published H3K9me2 data (Chen et al. *Genes Dev* 2012, Wilson et al. *Oncotarget* 2017).

We thank the reviewer for the suggestion and we now include H3K9me2 ChIP data from livers of fasted and refeed mice that showed H3K9me2 enrichment on lipogenic promoters in the fasted sample, in agreement with our concept that JMJD1C demethylates H3K9me2 upon refeeding for transcriptional activation of lipogenic genes. The TSS plots we present were generated using all aligned reads.

How ATAC-seq and ChIP-seq samples were analyzed needs to be included; were replicates performed-how consistent?, were samples pooled from more than one mouse, how many mice analyzed?, how were the peak differences quantified? Box plots and/or violin plots showing read density comparisons and statistics need to be include to assess the peak differences in the comparative data sets. Were scales in Genome browser shots the same? These are need to compare peak heights. RNA seq comparison in Fig. 5C needs more full description of the exact comparison being made.

As suggested by the reviewer, we edited the methods section to describe our sample analysis. Briefly, we had replicates for both ATAC-seq and ChIP-seq experiments and include analyses to show consistency between samples (Fig. S5A-B). Each replicate for both ATAC-seq and ChIP-seq experiments represented a separate mouse and we used two mice per condition individually. The replicate reads were grouped and processed together to align reads to mm10 genome assembly, determine peaks, gene lists, and pathway analysis. We now include box plots showing read density comparisons for each data set. Scales in the genome browser shots were the same to allow for comparison of the peak heights. We edited the description and specified the exact comparison made in Fig. 5C.

8). The dramatic difference in alpha-ketoglutarate levels is surprising and could have more profound implications for other JMJD family proteins as well as for TCA cycle flux and glutamate metabolism. How does the measurement of alpha-ketoglutarate levels compare with other reports in the literature for fast/feeding or even circadian variation where lipogenesis also varies?

Others reported that during short term fasting, the level of aKG decreases significantly (Minassian et al. J Bio Chem 1994), which is in agreement with our results. Interestingly, during prolonged fasting or starvation the level of aKG has been reported to increase back to basal levels (Minassian et al. J Bio Chem 1994), and the reason for this was explained by the authors as increase is “starvation based anaplerotic gluconeogenesis” (Wu et al. Biomol Ther 2016). As suggested by the reviewer, indeed, circadian variation of hepatic aKG and glutamine levels has been reported (Robinson et al. J Nutri 1981) with levels decreasing during the dark cycle and rebounding during the light cycle. In all our experiments, we collected tissue samples from fasted or fed mice at the same time of the day at noon, thus avoiding circadian variations. Circadian variation would not be a factor in our in vitro cell culture studies either.

9). JMJD1C protein expression loading control needed in Fig. 7C to ensure T505A is not expressed at lower level.

We thank the reviewer for pointing this omission out, we repeated the luciferase assay with the multiple JMJD1C mutants and now include loading controls showing similar protein levels among the WT and mutant samples to confirm that the changes we see are due to T505A mutation and not to lower protein levels.

10). References to “previously described” methods need to have references associated

Thank you. We corrected the manuscript and provide references.

Reviewer #2

In this manuscript, Viscarra et al. conducted a series of experiments to show that JMJD1C is recruited by USF-1 to various lipogenic genes for H3K9 demethylation to enhance chromatin accessibility in the fed state. This is achieved via mTORC1-mediated phosphorylation of JMJD1C on threonine 505 and increased availability of the co-substrate α -ketoglutarate in response to insulin or feeding.

Based on their observations, the authors infer that their “*results provide evidence that hepatosteatosis, rather than adipose expansion or the size of adipose tissue depots, may be more critical in the development of insulin resistance that accompanies diet-induced obesity*”.

The authors provide novel insights on regulatory mechanisms of hepatic lipogenesis that are relevant to hepatosteatosis. There are a few concerns that the authors should consider carefully.

We greatly appreciate positive comments of the reviewer.

Major concerns

1. H3K9me1 has been shown as histone mark of distinct regions of silent chromatin (*J Biol Chem* 2006 May 5;281(18):12760-6; *Biochimie* Vol 94 (12), December 2012, pp 2656-2664; *Experimental & Molecular Medicine* Vol 49, p e324 (2017); *Genes Dev.* 2002 Jul 15;16(14):1779-91.). The authors cite a paper suggesting that H4K9me1 is found in the promoter regions of active genes (*Cell* Vol 138, (5), 2009, pp 1019-1031). This controversy should be properly discussed. In any case, since HepG2 cells treated with insulin and mice refed after a fasting period show increased H3K9me1 levels on lipogenic genes whose expression increases under such conditions, how do the authors explain this observation? How can it be that the increase of a mark of silent chromatin leads to increased expression of lipogenic genes? Is it possible that, depending on the chromatin/promoter context, H3K9me1 may act either as mark of silent or active genes

From our own literature searches, as well as in the articles provided by the reviewer, the actual role of H3K9me1 whether activating or repressive mark remains unclear. While H3K9me3 is strongly correlated with silent chromatin, H3K9me1 appears to be more of a passive mark, not really demarcating silent or active genes. In the case of lipogenic promoters, we clearly observed H3K9me1 accumulation correlating with transcriptional activation. However, it is possible that accumulation of H3K9me1 was due to the fact that JMJD1C has higher demethylase activity for H3K9me2 than H3K9me1. Thus, H3K9me2 was demethylated by JMJD1C more efficiently than H3K9me1, resulting in H3K9me1 accumulation when lipogenic genes were activated. We include discussion on this controversy in the text.

Also, H3K4me3, a mark of active promoters increases under the same conditions. Is it possible that H3K4me3 is a stronger epigenetic mark than H3K9me1? The authors should provide evidences of how these two histone marks play a role in the expression of lipogenic genes. Moreover, how do the authors

explain that H3K9me1 levels increase under insulin or fed conditions when JMJD1C should demethylate H3K9me1?

We agree with the reviewer, in the literature, the H3K4me3 mark is strongly associated with active promoters. In Fig. 4 we show that H3K4me3 enrichment increases 3-5 fold upon insulin treatment and more than 6 fold upon refeeding, while H3K9me1 enrichment only increases about 2-3 fold under both conditions. Moreover, in Fig. 4C we also show that JMJD1C has very high specificity for H3K9me2 demethylation, as we detect no demethylation of H3K9me3, and only partial demethylation of H3K9me1 in our in vitro demethylation assays. Thus, under insulin or fed conditions, JMJD1C has higher demethylase activity towards H3K9me2, leading to H3K9me1 accumulation.

2. The quality of western blot in Figure 6A right is poor: the band of fasted mice total JMJD1C is covered with a white spot, likely due to uneven blotting and transfer in this part of the gel.

Per the reviewers suggestion, we repeated the experiment and have replaced the blot with one that doesn't obscure the fasted lane with a white spot.

3. Likewise, the WB images of co-IP experiment in Figure 7G lower panel is poor: no band are visible in input samples; also, the bands in the IP of USF-1 are quite faint! This experiment should be repeated to provide a more convincing image of the bands.

As suggested by the reviewer, we repeated this IP experiment and replaced the faint blot with one that clearly and convincingly showed each band in the input and IP lanes.

4. The authors claim that their "*results provide evidence that hepatosteatosis, rather than adipose expansion or the size of adipose tissue depots, may be more critical in the development of insulin resistance that accompanies diet-induced obesity*". I believe that this statement is a bit strong and I suggest to tone it down. A study with longer treatment with HFD should be performed to corroborate the view that hepatosteatosis, rather than adipose tissue expansion or size of adipose tissue depots, may be more critical in the development of insulin resistance following diet-induced obesity; with the present data it cannot be ruled out that adipose tissue expansion contributes to insulin resistance. It is likely that both hepatosteatosis and adipose tissue expansion contribute together, in a concerted and stepwise fashion, to the onset of insulin resistance. For instance, it cannot be ruled out that some signal originating from adipose tissue in mice treated with HFD (e.g., proinflammatory cytokines that are released by WAT) may trigger hepatosteatosis. At this stage of knowledge, I would be cautious to overstate the primary importance of hepatosteatosis and rule out the role of WAT in the onset of insulin resistance.

We now include data from WT and JMJD1C-LKO mice on chow diet for 4 months to compare with those on high CHO and high fat diet. WT and JMJD1C-LKO animals on chow diet remained insulin sensitive by GTT and ITT, although JMJD1C-LKO mice had lower hepatic TG.

Upon high fat diet feeding, both WT and JMJD1C-LKO mice showed higher adiposity. However, JMJD1C-LKO mice had lower hepatic TG levels and these KO mice were protected from obesity-related insulin resistance.

Upon high CHO diet feeding, JMJD1C-LKO mice had significantly lower liver TG levels compared WT mice. High CHO diet feeding caused insulin resistance in WT mice, whereas JMJD1C-LKO remained insulin sensitive, prevented from high CHO diet-induced insulin resistance. Importantly, neither WT nor KO mice were obese and they showed similar fat pad weights.

Therefore, JMJD1C ablation prevented high CHO diet-induced hepatosteatosis and insulin resistance in the absence of alterations in adiposity. Yet, as suggested by the reviewer, we toned down and made our point more specific. We now conclude that dysregulation of a specific histone demethylase, JMJD1C, causes hepatosteatosis and that hepatosteatosis even in the absence of altered adiposity can affect insulin resistance.

Minor concerns

1. Figure 3A middle panel shows results of JMJD1C gene expression in brain liver and kidney. The same results seem to be reported in Figure S2 panel A with the addition of JMJD1C gene expression in the lung, heart, spleen and muscle. I suggest to simply replace Figure 3A middle panel with Figure S2 panel A. Of course, Figure S2A should be removed from Figure S2.

We thank the reviewer for this suggestion, we have edited the figure so that the gene expression in various tissues is shown together in Figure 3.

2. Pg 11, line 226: the sentence “*demonstrating that JMJD1C protected these mice from hepatosteatosi*s” should be corrected as follows: “*demonstrating that knock out of JMJD1C in the liver protected these mice from hepatosteatosi*s”. Also, the following sentence “*Glycogen levels upon refeeding were again significantly higher in the JMJD1C-LKO livers, compared to WT mice (Fig. 3B bottom left)*” should be “*Glycogen levels upon refeeding were again significantly higher in the JMJD1C-LKO livers, compared to WT mice (Fig. 3B middle right)*”.

We thank the reviewer for catching these errors, we have edited the text as suggested.

3. Figure S2C left panel: images of different fat depots are not labelled. Presumably, as stated in the figure legend, subWAT are the images right below the liver images, epi-WAT the next ones and BAT the images at the bottom of the panel, however I recommend labelling the images directly in the panel. Further, in the right panel the quantification of WAT refers to subWAT, epi-WAT or to the sum of both? Please, specify.

As suggested by the reviewer, we have edited the figure to include labels next to the tissue pictures. The quantification of WAT refers to the sum of both, we have edited the figure legend to explain this.

Reviewer #3

In this original work submission Viscarra and colleagues characterize the physiologic role and mechanisms of the histone demethylase JMJD1C in hepatic insulin responses. The authors postulate that JMJD1C undergoes site-specific phosphorylation by mTOR and is recruited to lipogenic gene promoters in response to fast/refeed. They further surmise that JMJD1C demethylates specific histone marks to alter chromatin dynamics at lipogenic genes.

The work utilizes unbiased approaches that build logically on multiple previous publications from this group and adds to our understanding of mechanisms orchestrating promoter activation in metabolic control. The text is well-written although figure renderings could be improved. There are a number of issues with the proposed mechanism and other technical considerations that should be addressed but on balance this is a novel and timely story.

We thank the reviewer for their positive comments.

1. The authors state that collaborative interaction of transcription factors at lipogenic promoters in response to insulin require an accessible chromatin landscape.

One would presume then that overexpression of nuclear SREBP1c may not rescue the phenotype of JMJD1C liver specific knockout since chromatin architecture may be limiting (would be a highly striking finding). Is this the case? The authors should administer control virus and AAV (or adeno) SREBP1c in WT and L-KO to test this directly. Although convincing evidence is presented that JMJD1C contributes to epigenetic landscape and transcriptional outputs at lipogenic genes, the epistatic relationship between JMJD1C and canonical transcriptional regulators in insulin responses is not established and these experiments will provide critical insight.

According to the reviewer's suggestion, we used our JMJD1C-LKO mice to document the effects of SREBP-1c overexpression as well as LXR agonist treatment to compare with the effects on WT mice. As expected, overexpression of SREBP-1c activated lipogenic genes, whereas in JMJD1C-LKO mice, the degree of induction of lipogenic genes was significantly decreased. Observed blunted effects of SREBP-1c overexpression probably reflects the above new data that show interaction of JMJD1C with SREBP-1c albeit indirect (direct interaction with USF-1). It is also possible that JMJD1C depletion prevented chromatin accessibility at lipogenic promoters as shown in our ATAC-seq, thus SREBP-1c could not function well.

2. Lipogenic genes are regulated in response to dietary clues through diverse mechanisms but the notion that JMJD1C induces lipogenic genes ONLY in response to fast-refeed is not convincingly established. The nuclear receptors LXRs are direct inducers of lipogenesis in response to cholesterol overload in addition to contributing to fast/refeed responses. Does the administration of LXR agonists (readily available commercially) in the setting of JMJD1C loss of function show differential regulation of lipogenic genes? How about administration of cholesterol rich diet? The studies will clarify the specificity of JMJD1C effects on lipogenic genes.

Although we are focusing on lipogenic gene induction in response to insulin/feeding and here we described JMJD1C recruitment by USF-1 and JMJD1C phosphorylation by mTORC, we agree with the reviewer that lipogenic genes are regulated in response to diverse dietary cues. LXR can directly bind upstream region of FAS promoter for transcriptional activation as well as indirectly by inducing SREBP-1c. Thus, according to the reviewer's suggestion, we performed experiments using LXR agonist in WT and JMJD1C LKO mice. As expected, LXR agonist increased lipogenic gene transcription in WT mice, probably via direct as well as indirect mechanism involving SREBP1c (Fig. S3B). In JMJD1C-KO mice, we found the lipogenic gene induction by LXR agonist blunted, albeit partially. This could be due to indirect interaction with JMJD1C described below (Figure S1) and/or by condensed chromatin due to lack of JMJD1C ablation as we shown in our ATAC-seq (Fig. 5),

It has been well documented that cholesterol overload decreases lipogenic gene expression, while increasing genes in cholesterol elimination (Tsai et al. J Nutri 1975, Bennett et.al. J Biol Chem 1995, Shimomura et al. PNAS 1997, Wang et al. Lipi Health Dis 2010). Regardless, as suggested by the reviewer, we fed WT and JMJD1C-LKO mice with high cholesterol diet. In WT mice, as expected, lipogenic gene expression was decreased by 20% upon cholesterol compared to chow feeding. JMJD1C-LKO mice on chow diet already had low lipogenic gene expression, making it impossible to study suppressive effect of cholesterol feeding (Fig. S3).

We addressed whether JMJD1C can directly interact with other transcription factors known to regulate lipogenesis, such as ChREBP, SREBP-1c or LXR. Co-IP of lysates from 293FT cells overexpressing SREBP-1c, or LXR detected interaction of JMJD1C with SREBP-1c and LXR but not with ChREBP (Fig. S1A). Our pull-down of recombinant JMJD1C with purified SREBP-1c or LXR did not detect interaction of JMJD1C with SREBP-1c or LXR (Fig. S1B). These results indicate that, although SREBP-1c or LXR can be found in JMJD1C complex, JMJD1C does not directly interact with SREBP-1c or LXR. In thus regard, we previously reported that USF-1 directly interacts with SREBP1 to recruit it to the promoter regions of lipogenic genes (Griffin et al. J Bio Chem 2007, Latasa et al, PNAS, 2000) and thus via direct interaction with USF-1, SREBP-1c could be found in complex containing JMJD1C. As for LXR, further studies would be needed to understand how LXR can indirectly interact with JMJD1C.

In this revision, we include and discuss these additional results, to provide a more general picture how JMJD1C functions in lipogenic gene activation.

3. Insulin and fast refeed responses activate lipogenic genes in addition to inhibiting gluconeogenesis (PMID 20133650). Does loss of JMJD1C influence the expression of gluconeogenic genes in response to insulin and fast/refeed? The authors should show clear qPCR analysis since the ATAC-seq of gluconeogenic genes in S3 are difficult to interpret (based on in vivo data it appears that JMJD1C strongly influences glucose responses and figure 4C suggests that JMJD1C may perhaps have additional effects on chromatin besides demethylating H3K9me2 and 1). If JMJD1C does influence gluconeogenesis how do the authors consolidate previous findings showing that insulin suppression of gluconeogenesis does NOT require mTOR? Should be addressed in discussion

In our ATAC-seq of fasting vs feeding, gluconeogenic genes were in open chromatin in fasted condition, while closed in fed condition, and in our ChIP-seq of H3K9me2 of fasting vs feeding, H3K9me2 were enriched in gluconeogenic genes in fed condition, which all correspond to their well-recognized activation in fasting and repression in fed condition. However, we did not detect any differences in gluconeogenic genes in RNA-seq and ATAC-seq comparing JMJD1C-LKO vs WT mice. Even so, according to the reviewer's suggestion, we also performed RT-qPCR and found no changes in Pck1, G6pc, and Gk, as well as PGC1a in JMJD1C-LKO compared WT liver (Fig. S4), further demonstrating specific JMJD1C function in lipogenic gene activation (We note that, as expected and recognized, some of the glycolytic enzymes are activated in fed condition in a similar manner to lipogenesis). There are numerous H3K9 methylases and demethylases other than JMJD1C, some of those could be involved in regulation of gluconeogenic genes. As suggested by the reviewer, this discussion is included now.

4. Did the authors perform genome-wide normalization for ATAC-seq data for L- JMJD1C KO? The background appears to be much higher in L- JMJD1C KO samples so would be cautious in making the claim that access sites show differential openness. Did the authors perform motif analysis on differential comparing WT and L- JMJD1C KO refeed?

We thank the reviewer for this suggestion, we repeated our analyses employing genome wide normalization, which removed a lot of background, and have amended our results and discussion accordingly. The majority of our detected peaks still appear to be centered around the TSS (Fig. 5). We performed motif analysis as suggested searching for differential enrichment of known motifs and detected 55 motifs which lost significant enrichment in the JMJD1C-LKO samples. Of note is the motif for USF-1 which goes from a p-value of 9.52E-08 in the WT to 0.77 in the KO (Table S2).

Minor

1. Please provide more details on ATAC seq analysis. For example what fold cutoff was used for accessible peaks or did you use a specific statistical test?

As suggested by the reviewer, we have edited the methods section to include thorough descriptions of our ATAC-seq and ChIP-seq analyses. For initial peak detection we used MACS2 in ATAC mode with a q-value cutoff of 0.05 and fold enrichment cutoff of 2.0. We then performed ANOVA to identify peaks that changed significantly in either our Fasted to Refed, or WT to JMJD1C-LKO comparisons.

2. The authors should show protein blots for a number of lipogenic genes related to figures 2 and 3. It appears that changes in gene expression are substantially more dramatic than observed phenotype.

As requested by the reviewer, we have added blots for FAS and SREBP-1c to go along with the JMJD1C overexpression and knockout gene expression data provided (Fig. 2 and 3).

3. The genetic background for the JMJD1C flox mice is presumably bl6. Please confirm in methods since this is not clear from the checklist.

As suggested by the reviewer, we changed the methods section to confirm all mice are C57/BL6.

4. The figures and figure legends are frustratingly difficult to follow with excessive use of right, left, middle top etc... For many figure groupings break the panels into individual figures using additional letters. For example, see panels figure 2A, 2C, and 2D.

We agree with the reviewer. We have changed the labeling of figures as suggested to make them easier to follow by breaking groupings with too many components, for example, Fig.3 previously just had A and B, but we broke the panels apart from A to N making the figure legends more clear.

5. The paper relies heavily on 293 and HepG2 cells which do not entirely recapitulate in vivo responses and for this reason not used extensively in previous investigations of hepatic insulin responses. Suggest discussing in limitations.

In addition to the cell culture systems, we performed extensive in vivo studies. We agree with the reviewer and we have indicated limitations of using cultured cells in the text.

REVIEWERS' COMMENTS:

Reviewer #1 (Remarks to the Author):

The authors have included a significant amount of new experimental data and have modified the manuscript extensively. These modifications have greatly increased the quality of the manuscript. However, the box plots in Figure S5B are not explained, whether there is a significant difference is not indicated and how this information addresses the question of sample consistency as suggested by the authors is not clear.

Reviewer #2 (Remarks to the Author):

In the revised version of the manuscript NCOMMS-19-12666A, authors addressed all major and minor concerns that were raised upon the submission of the first version of this manuscript. I have no further comments.

Reviewer #3 (Remarks to the Author):

The authors have addressed the vast majority of concerns and I am supportive of publication.

Response to reviewers

REVIEWERS' COMMENTS:

Reviewer #1 (Remarks to the Author):

The authors have included a significant amount of new experimental data and have modified the manuscript extensively. These modifications have greatly increased the quality of the manuscript. However, the box plots in Figure S5B are not explained, whether there is a significant difference is not indicated and how this information addresses the question of sample consistency as suggested by the authors is not clear.

We have updated the figure legend for Supplementary Figure 5A and B to describe that the box plots presented show the number and distribution of reads between groups. Number and distribution of reads of the different groups are similar and thus the datasets are able to be compared. Furthermore, we now include a statistical analysis section in the methods to describe in more detail the analyses we performed for genome wide datasets. As stated in the methods, we use ANOVA to compare the datasets to determine significant changes between the groups as well as using pathway analysis to highlight significantly different biological processes between groups. These significantly different processes are the ones presented in Figure 5 of the main text.

Reviewer #2 (Remarks to the Author):

In the revised version of the manuscript NCOMMS-19-12666A, authors addressed all major and minor concerns that were raised upon the submission of the first version of this manuscript. I have no further comments.

Reviewer #3 (Remarks to the Author):

The authors have addressed the vast majority of concerns and I am supportive of publication.